# Improving VAEs' Robustness to Adversarial Attack

**Matthew Willetts**[*,1,2]   **Alexander Camuto**[*,1,2]   **Tom Rainforth**[1]
**Stephen Roberts**[1,2]   **Chris Holmes**[1,2]
[1]University of Oxford   [2]Alan Turing Institute, London

## ABSTRACT

Variational autoencoders (VAEs) have recently been shown to be vulnerable to adversarial attacks, wherein they are fooled into reconstructing a chosen target image. However, how to defend against such attacks remains an open problem. We make significant advances in addressing this issue by introducing methods for producing adversarially robust VAEs. Namely, we first demonstrate that methods proposed to obtain disentangled latent representations produce VAEs that are more robust to these attacks. However, this robustness comes at the cost of reducing the quality of the reconstructions. We ameliorate this by applying disentangling methods to hierarchical VAEs. The resulting models produce high–fidelity autoencoders that are also adversarially robust. We confirm their capabilities on several different datasets and with current state–of–the–art VAE adversarial attacks, and also show that they increase the robustness of downstream tasks to attack.

## 1 INTRODUCTION

Variational autoencoders (VAEs) are a powerful approach to learning deep generative models and probabilistic autoencoders (Kingma & Welling, 2014; Rezende et al., 2014). However, previous work has shown that they are vulnerable to adversarial attacks (Tabacof et al., 2016; Gondim-Ribeiro et al., 2018; Kos et al., 2018): an adversary attempts to fool the VAE to produce reconstructions similar to a chosen target by adding distortions to the original input, as shown in Fig 1. This kind of attack can be harmful when the encoder's output is used downstream, as in Xu et al. (2017); Kusner et al. (2017); Theis et al. (2017); Townsend et al. (2019); Ha & Schmidhuber (2018); Higgins et al. (2017b). As VAEs are often themselves used to protect classifiers from adversarial attack (Schott et al., 2019; Ghosh et al., 2019), ensuring VAEs are robust to adversarial attack is an important endeavour.

Despite these vulnerabilities, little progress has been made in the literature on how to ***defend*** VAEs from such attacks. The aim of this paper is to investigate and introduce possible strategies for defence. We seek to defend VAEs in a manner that maintains reconstruction performance. Further, we are also interested in whether methods for defence increase the robustness of downstream tasks using VAEs.

Our first contribution is to show that regularising the variational objective during training can lead to more robust VAEs. Specifically, we leverage ideas from the disentanglement literature (Mathieu et al., 2019) to improve VAEs' robustness by learning smoother, more stochastic representations that are less vulnerable to attack. In particular, we show that the total correlation (TC) term used to encourage independence between latents of the learned representations (Kim & Mnih, 2018; Chen et al., 2018; Esmaeili et al., 2019) also serves as an effective regulariser for learning robust VAEs.

Though a clear improvement over the standard VAE, a severe drawback of this approach is that the gains in robustness are coupled with drops in the reconstruction performance, due to the increased regularisation. Furthermore, we find that the achievable robustness with this approach can be limited (see Fig 1) and thus potentially insufficient for particularly sensitive tasks. To address this, we apply TC–regularisation to *hierarchical* VAEs. By using a richer latent space representation than a standard VAE, the resulting models are not only more robust still to adversarial attacks than single-layer models with TC regularisation, but can also provide reconstructions which are comparable to, and often even better than, the standard (unregularised, single-layer) VAE.

---

[*]Equal Contribution. Contact at: `mwilletts@turing.ac.uk`; `acamuto@turing.ac.uk`

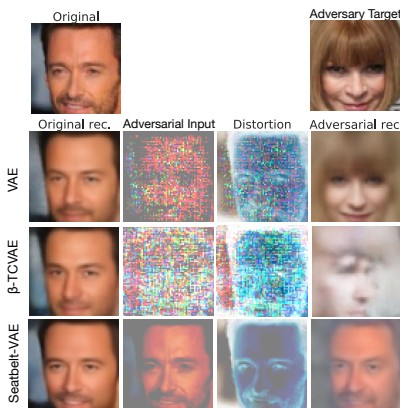

Figure 1: Adversarial attacks on CelebA for different models. Here we start with the image of Hugh Jackman and introduce an adversary that tries to produce reconstructions that look like Anna Wintour. This is done by applying a distortion (third column) to the original image to produce an adversarial input (second column). We can see that the adversarial reconstruction for the Vanilla VAE looks substantially like Wintour, indicating a successful attack. Adding a regularisation term using the $\beta$-TCVAE produces an adversarial reconstruction that does not look like Wintour, but it is also far from a successful reconstruction. The hierarchical version of a $\beta$-TCVAE (which we call Seatbelt-VAE) is sufficiently hard to attack that the output under attack still looks like Jackman, not Wintour.

To summarise: We provide insights into what makes VAEs vulnerable to attack and how we might go about defending them. We unearth novel connections between disentanglement and adversarial robustness. We demonstrate that regularised VAEs, trained with an up-weighted total correlation, are much more robust to attacks than vanilla VAEs. Building on this we develop regularised hierarchical VAEs that are more robustness still and offer improved reconstructions. Finally, we show that robustness to adversarial attack also confers increased robustness to downstream tasks.

## 2 BACKGROUND: ATTACKING VAEs

In adversarial attacks an agent is trying to manipulate the behaviour of some model towards a goal of their choosing (Akhtar & Mian, 2018; Gilmer et al., 2018). For many deep learning models, very small changes in the input can produce large changes in output. Attacks on VAEs have been proposed where the adversary looks to apply small input distortions that produce reconstructions close to a target adversarial image (Tabacof et al., 2016; Gondim-Ribeiro et al., 2018; Kos et al., 2018). An example is shown in Fig 1: a standard VAE is successfully attacked, turning Jackman into Wintour.

Unlike more established adversarial settings, only a small number of such VAE attacks have been suggested in the literature. The current known most effective mode of attack is a *latent space attack* (Tabacof et al., 2016; Gondim-Ribeiro et al., 2018; Kos et al., 2018). This aims to find a distorted image $\mathbf{x}^* = \mathbf{x} + \mathbf{d}$ such that its posterior $q_\phi(\mathbf{z}|\mathbf{x}^*)$ is close to that of the agent's chosen target image $q_\phi(\mathbf{z}|\mathbf{x}^t)$ under some metric. This then implies that the likelihood $p_\theta(\mathbf{x}^t|\mathbf{z})$ is high when given draws from the posterior of the adversarial example. It is particularly important to be robust to this attack if one is concerned with using the encoder network of a VAE as part of a downstream task. For a VAE with a single stochastic layer, the latent-space adversarial objective is

$$\Delta_{\mathrm{r}}(\mathbf{x}, \mathbf{d}, \mathbf{x}^t; \lambda) = r(q_\phi(\mathbf{z}|\mathbf{x} + \mathbf{d}), q_\phi(\mathbf{z}|\mathbf{x}^t)) + \lambda||\mathbf{d}||_2, \tag{1}$$

where $r(\cdot, \cdot)$ is some divergence or distance, commonly a $D_{\mathrm{KL}}$(Tabacof et al., 2016; Gondim-Ribeiro et al., 2018). We are penalising the $L_2$ norm of $\mathbf{d}$ too, so as to aim for attacks that change the image less. We can then simply optimise to find a good distortion $\mathbf{d}$.

Alternatively, we can aim to directly increase the ELBO for the target datapoint (Kos et al., 2018):

$$\Delta_{\mathrm{output}}(\mathbf{x}, \mathbf{d}, \mathbf{x}^t; \lambda) = \mathbb{E}_{q_\phi(\mathbf{z}|\mathbf{x}+\mathbf{d})}\left[\log(\mathbf{x}^t|\mathbf{z})\right] - D_{\mathrm{KL}}(q_\phi(\mathbf{z}|\mathbf{x} + \mathbf{d})||p(\mathbf{z})) + \lambda||\mathbf{d}||_2. \tag{2}$$

## 3 DEFENDING VAEs

This problem was not considered by prior works[1]. To address it, we first need to consider what makes VAEs vulnerable to adversarial attacks. We argue that two key factors dictate whether we can perform a successful attack on a VAE: a) whether we can induce significant changes in the encoding distribution $q_\phi(\mathbf{z}|\mathbf{x})$ through only small changes in the data $\mathbf{x}$, and b) whether we can induce significant changes in the reconstructed images through only small changes to the latents $\mathbf{z}$. The first of these relates to the *smoothness* of the encoder mapping, the latter to the smoothness of the decoder mapping.

---

[1]We note that the earliest version of this work appeared in June 2019 (Willetts et al., 2019), here extended. Since then other works, eg Camuto et al. (2020); Cemgil et al. (2020); Barrett et al. (2021), have built of our own to consider this problem of VAE robustness, including investigating it from a more theoretical standpoint.

Consider, for the sake of argument, the case where the encoder–decoder process is almost completely noiseless. Here successful reconstruction places no direct pressure for similar encodings to correspond to similar images: given sufficiently powerful networks, very small changes to embeddings $\mathbf{z}$ can imply very large changes to the reconstructed image; there is no ambiguity in the "correct" encoding of a particular datapoint. In essence, we can have lookup–table style behaviour – nearby realisations of $\mathbf{z}$ do not necessarily relate to each other and very different images can have very similar encodings.

This will now be very vulnerable to adversarial attacks: small input changes can lead to large changes in the encoding, and small encoding changes can lead to large changes in the reconstruction. It will also tend to overfit and have gaps in the aggregate posterior, $q_\phi(\mathbf{z}) = \frac{1}{N} \sum_{n=1}^{N} q_\phi(\mathbf{z}|\mathbf{x}_n)$, as each $q_\phi(\mathbf{z}|\mathbf{x}_n)$ will be sharply peaked. These gaps can then be exploited by an adversary.

There are two mechanisms by which we can reduce this lookup-table behaviour, thereby reducing gaps in the aggregate posterior. First, we can try to regulate the level of noise in the per-datapoint posterior covariance, to then obtain smoothness in the overall embeddings. Having a stochastic encoding creates uncertainty in the latent that gives rise to a particular image, forcing similar latents to correspond to similar images. Adding noise forces the VAE to smooth the encode-decode process in that similar images will lead to similar embeddings in the latent space, ensuring that small changes in the input result in small changes in the latent space and result in small changes in the decoded outputs. This proportional input-output change is what we refer to as a 'simple' encode-decode process, which is the second mechanism that can reduce look-up table behaviour.

The fact that the VAE is vulnerable to adversarial attack suggests that its standard setup does not obtain sufficiently smooth and simple representations to provide an adequate defence. Introducing additional regularisation to enforce simplicity or increased posterior covariance thus provides a prospect for defending VAEs. We could attempt to obtain this by direct regularisation of the networks (e.g. weight decay). Here, however, we focus on macro-level regularisation approaches as discussed in the next section. The reason for this is that controlling the macroscopic behaviour of the networks through low-level regularisations can be difficult to control and, in particular, difficult to calibrate. Further, as the most effective attack on VAEs currently attack the latent space, it is reasonable that regularisation methods that directly act on the properties of the latent space form a good place to start.

## 3.1 Disentangling Methods and Robustness

Recent research into disentangling VAEs (Higgins et al., 2017a; Siddharth et al., 2017; Kim & Mnih, 2018; Chen et al., 2018; Esmaeili et al., 2019; Mathieu et al., 2019) and the information bottleneck (Alemi et al., 2017; 2018) has looked to regularise the ELBO with the hope of providing more interpretable embeddings. These regularisers also have influences on the smoothness and stochasticity of the embeddings learned.

Of particular relevance, Mathieu et al. (2019) introduce the notion of ***overlap*** in the embedding of a VAE: the level of overlap between per-datapoint posteriors as they combine to form the aggregate posterior. Controlling this is critical to achieving a smoothly varying latent embedding. Overlap encapsulates both the level of uncertainty in the encoding process and also a locality of this uncertainty. To learn a smooth representation we not only need our encoder distribution to have an appropriate entropy, we also want the different possible encodings to be similar to each other. Critically, Mathieu et al. (2019) show that many methods proposed for disentangling, and in particular the $\beta$-VAE (Higgins et al., 2017a; Alemi et al., 2017), provide a mechanism for directly controlling this overlap.

Going back to our previous arguments, we see that controlling this overlap may also provide a mechanism for improving VAEs' robustness. This observation now hints at an interesting question: *can we use methods initially proposed to encourage disentanglement to encourage robustness*?

It is important to note here that disentangling can be difficult to achieve in practice, typically requiring precise choices in the hyperparameters of the model and the weighting of the added regularisation term, and often also a fair degree of luck (Locatello et al., 2019; Mathieu et al., 2019; Rolinek et al., 2019). As such, we are not suggesting to induce *disentangled representations* to induces robustness, or indeed that disentangled representations should be any more robust. Rather, as highlighted above, we are interested in whether the regularisers traditionally used to encourage disentanglement reliably lead to adversarially robust VAEs. Indeed, we will find that though our approaches—based on these regularisers—provide reliable and significant improvements in robustness, these improvements are not generally due to any noticeable improvements in disentanglement itself (see Appendix E.1).

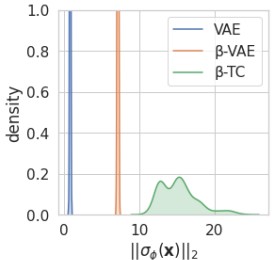 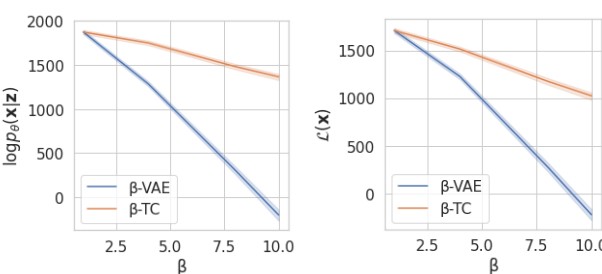

Figure 2: [Left] density plot of $||\boldsymbol{\sigma}_\phi(\mathbf{x})||_2$ (the norm of the encoder standard deviation) for a VAE, a $\beta$-VAE and a $\beta$-TCVAE each trained on CelebA, $\beta = 10$. The $\beta$-VAE's posterior variance saturates, while the $\beta$-TCVAE's does not and as such is able to induce more overlap. [Right] the likelihood ($\log p_\theta(\mathbf{x}|\mathbf{z})$) and ELBO for both as a function of $\beta$. Clearly the model quality degrades to a lesser degree for the TC-penalised models under increasing $\beta$.

**Regularising for Robustness**  There are a number of different disentanglement methods that one might consider using to train robust VAEs. Perhaps the simplest would be to use a $\beta$-VAE (Higgins et al., 2017a), wherein we up-weight the $D_{\mathrm{KL}}$ term in the VAE's ELBO by a factor $\beta \geq 1$. However, as mentioned previously the $\beta$-VAE only increases overlap at the expense of substantial reductions in reconstruction quality as the data likelihood term has, in effect, been down-weighted (Kim & Mnih, 2018; Chen et al., 2018; Mathieu et al., 2019).

Because of these shortfalls, we instead propose to regularise through penalisation of a total correlation (TC) term (Kim & Mnih, 2018; Chen et al., 2018). As discussed in Section A.1, this looks to directly force independence across the different latent dimensions in aggregate posterior $q_\phi(\mathbf{z})$, such that the aggregate posterior factorises across dimensions. This approach has been shown to have a smaller deleterious effect on reconstruction quality than found in $\beta$-VAEs (Chen et al., 2018). As seen in Fig 2 this method also gives greater overlap by increasing posterior variance. To summarise, the greater overlap and the lesser degradation of reconstruction quality induced by $\beta$-TCVAE make them highly suitable for our purposes.

## 3.2    Adversarial Attacks on TC-Penalised VAEs

We now consider attacking these TC-penalised VAEs and demonstrate one of the key contributions of the paper: that empirically this form of regularisation makes adversarial attacks on VAEs harder to carry out. To do this, we first train them under the $\beta$-TCVAE objective (i.e. Eq (15)), jointly optimising $\theta, \phi$ for a given $\beta$. Once trained, we then attack the models using the latent-space attack method outlined in Section 2, finding an input distortion $\mathbf{d}$ that minimises the latent attack loss $\Delta$ as per Eq (1) with $r(\cdot, \cdot) = D_{\mathrm{KL}}(\cdot||\cdot)$.

One possible metric for how successful such attacks have been is the achieved value reached of the attack loss $\Delta_{\mathrm{KL}}$. If the latent space distributions for the original input and for the distorted input match closely for a small distortion, then $\Delta_{\mathrm{KL}}$ is small and the model has been successfully fooled – reconstructions from samples from the attacked posterior would be indistinguishable from those from the target posterior. Meanwhile, the larger the converged value of the attack loss the less similar these distributions are and the more different the reconstructed image is to the adversarial target image.

We carry our these attacks for dSprites (Matthey et al., 2017), Chairs (Aubry et al., 2014) and 3D faces (Paysan et al., 2009), for a range of $\beta$ and $\lambda$ values. We pick values of $\lambda$ following standard methodology (Tabacof et al., 2016; Gondim-Ribeiro et al., 2018), and use L-BFGS-B for gradient descent (Byrd et al., 1995). We also varied the dimensionality of the latent space of the model, $d_{\mathbf{z}}$, but found it had little effect on the effectiveness of the attack.

In Fig 3 we show the effect on the attack loss $\Delta_{\mathrm{KL}}$ for varying $\beta$, averaged over different original input-target pairs and values of $d_{\mathbf{z}}$. Note that the plot is logarithmic in the loss. We see a clear pattern for each dataset that the loss values reached by the adversary increases as we increase $\beta$ from the standard VAE (i.e. $\beta = 1$). This analysis is also borne out by visual inspection of the effectiveness of these attacks, for example as shown in Fig 1. We will return to give further experimental results in Section 5. An interesting aspect of Fig 3 is that in many cases the adversarial loss starts to decrease if $\beta$ is too large: as $\beta$ increases there is less pressure in the objective to produce good reconstructions.

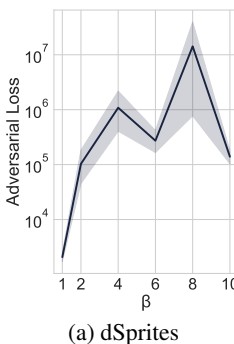 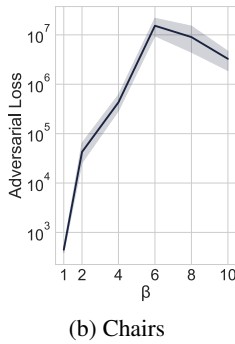 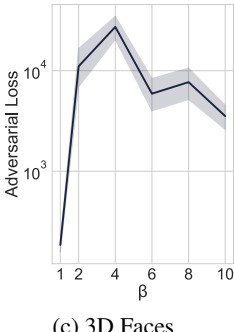

(a) dSprites  (b) Chairs  (c) 3D Faces

Figure 3: Attacker's achieved loss $\Delta_{\mathrm{KL}}$ (i.e. Eq (1) with $r = D_{\mathrm{KL}}$) for $\beta$-TCVAE for different $\beta$ values and datasets. Higher loss indicates more robustness. Shading corresponds to the 95% CI produced by attacking 20 images for each combination of $d_{\mathbf{z}} = \{4, 8, 16, 32, 64, 128\}$ and taking 50 geometrically distributed values of $\lambda$ between $2^{-20}$ and $2^{20}$ (giving 1000 total trials). Note that the loss axis is logarithmic. $\beta > 1$ clearly induces a much larger loss for the adversary relative to $\beta = 1$ for all datasets.

## 4 HIERARCHICAL $TC$–PENALISED VAEs

We are now armed with the fact that penalising the TC in the ELBO induces robustness in VAEs. However, TC-penalisation in single layer VAEs comes at the expense of model reconstruction quality (Chen et al., 2018), albeit less than that in $\beta$-VAEs. Our aim is to develop a model that is robust to adversarial attack while mitigating this trade-off between robustness and sample quality. To achieve this, we now consider instead using hierarchical VAEs (Rezende et al., 2014; Sønderby et al., 2016; Kingma et al., 2016; Zhao et al., 2017; Maaløe et al., 2019; Vahdat & Kautz, 2020; Child, 2021). These are known for their superior modelling capabilities and more accurate reconstructions. As these gains stem from using more complex hierarchical latent spaces, rather than less noisy encoders, this suggests they may be able to produce better reconstructions and generative capabilities, while also remaining robust to adversarial attacks when appropriately regularised.

The simplest hierarchical extension of conditional stochastic variables in the generative model is the Deep Latent Gaussian Model (DLGM) of Rezende et al. (2014). Here the forward model factorises as a chain, $p_\theta(\mathbf{x}, \vec{\mathbf{z}}) = p_\theta(\mathbf{x}|\mathbf{z}^1) \prod_{i=1}^{L-1} p_\theta(\mathbf{z}^i|\mathbf{z}^{i+1}) p(\mathbf{z}^L)$, where each $p_\theta(\mathbf{z}^i|\mathbf{z}^{i+1})$ is a Gaussian distribution with mean and variance parameterised by deep nets, while $p(\mathbf{z}^L)$ is an isotropic Gaussian. Unfortunately, we found that naively applying TC-correlation penalisation to DLGM-style VAEs did not confer the improved robustness we observed in single layer VAEs. We postulate that this observed weakness is inherent to the structure of chain factorisation in the generative model. This means that the data-likelihood depends solely on $\mathbf{z}^1$, the bottom-most latent variable, and attackers only need to manipulate $\mathbf{z}^1$ to produce a successful attack.

To account for this, we instead use a generative model in which the likelihood $p_\theta(\mathbf{x}|\vec{\mathbf{z}})$ depends on *all* the latent variables in the chain $\vec{\mathbf{z}}$, rather than just the bottom layer $\mathbf{z}^1$, as has been done in Kingma et al. (2016); Maaløe et al. (2019). This leads to the following factorisation of the generative structure:

$$p_\theta(\mathbf{x}, \vec{\mathbf{z}}) = p_\theta(\mathbf{x}|\vec{\mathbf{z}}) \prod_{i=1}^{L-1} p_\theta(\mathbf{z}^i|\mathbf{z}^{i+1}) p(\mathbf{z}^L). \tag{3}$$

To construct the ELBO, we must further introduce an inference network $q_\phi(\vec{\mathbf{z}}|\mathbf{x})$. On the basis of simplicity and that it produces effective empirical performance, we use the factorisation:

$$q_\phi(\vec{\mathbf{z}}|\mathbf{x}) = q_\phi(\mathbf{z}^1|\mathbf{x}) \prod_{i=1}^{L-1} q_\phi(\mathbf{z}^{i+1}|\mathbf{z}^i, \mathbf{x}), \tag{4}$$

where each conditional distribution $q_\phi(\mathbf{z}^{i+1}|\mathbf{z}^i, \mathbf{x})$ takes the form of a Gaussian. Again, marginalising out intermediate $\mathbf{z}^i$ layers, $q_\phi(\mathbf{z}^L|\mathbf{x})$ is a non-Gaussian, highly flexible distribution. To defend this model against adversarial attack, we apply TC regularisation term as per the last section. We refer to the resulting models as Seatbelt-VAEs. We obtain a decomposition of the ELBO for this model, revealing the existence of a TC term for the top-most layer (see Appendix B for proof).

**Theorem 1.** *The Evidence Lower Bound, for a hierarchical VAE with forward model as in Eq* (3) *and amortised variational posterior as in Eq* (4)*, can be decomposed to reveal the total correlation (see Definition A.1), of the aggregate posterior of the top-most layer of latent variables:*

$$\mathcal{L}(\theta, \phi; \mathcal{D}) = \mathbb{E}_{q(\vec{\mathbf{z}}, \mathbf{x})} \log p_\theta(\mathbf{x}|\vec{\mathbf{z}}) + \boxed{R} + \boxed{S_a} + \boxed{S_b} - D_{\mathrm{KL}}\Big(q(\mathbf{z}^L)|| \prod_j q(z_j^L)\Big), \quad (5)$$

*where the last term is the required TC term, and, using $j$ to index over the coordinates in $\mathbf{z}^L$,*

$$\boxed{R} = \int \mathrm{d}\mathbf{x} \prod_{i=1}^{L} (\mathrm{d}\mathbf{z}^i) q_\phi(\vec{\mathbf{z}}|\mathbf{x}) q(\mathbf{x}) \log \frac{\prod_{k=1}^{L-1} p_\theta(\mathbf{z}^k|\mathbf{z}^{k+1})}{q_\phi(\mathbf{z}^1|\mathbf{x}) \prod_{m=1}^{L-2} q_\phi(\mathbf{z}^{m+1}|\mathbf{z}^m, \mathbf{x})} \quad (6)$$

$$\boxed{S_a} = -\mathbb{E}_{q_\phi(\mathbf{z}^{L-1})} D_{\mathrm{KL}}(q_\phi(\mathbf{z}^L, \mathbf{x}|\mathbf{z}^{L-1})||q_\phi(\mathbf{z}^L)q(\mathbf{x})) \quad (7)$$

$$\boxed{S_b} = -\sum_j D_{\mathrm{KL}}(q_\phi(\mathbf{z}_j^L)||p(\mathbf{z}_j^L)). \quad (8)$$

In other words, following the Factor and $\beta$-TCVAEs, we up-weight the TC term for $\mathbf{z}^L$. We can upweight this term then recombine the decomposed parts of the ELBO, to give us the following compact form of this objective.

**Definition 1.** *A Seatbelt-VAE is a hierarchical VAE with forward model as in Eq* (3) *and amortised variational posterior as in Eq* (4)*, trained wrt its parameters $\theta, \phi$ to maximise the objective:*

$$\mathcal{L}^{\mathrm{Seatbelt}}(\theta, \phi; \beta, \mathcal{D}) := \mathbb{E}_{q_\phi(\vec{\mathbf{z}}, \mathbf{x})}\left[\log \frac{p_\theta(\mathbf{x}, \vec{\mathbf{z}})}{q_\phi(\vec{\mathbf{z}}|\mathbf{x})}\right] - (\beta - 1)D_{\mathrm{KL}}\Big(q(\mathbf{z}^L)||\prod_j q(z_j^L)\Big). \quad (9)$$

We see that, when $L = 1$, a Seatbelt-VAE reduces to a $\beta$-TCVAE. We use the $\beta = 1$ case as a baseline in our experiments as it corresponds to a Vanilla VAE for $L = 1$ and for $L > 1, \beta = 1$ it produces a hierarchical model with a likelihood function conditioned on all latents.

As with the $\beta$-TCVAE, training $\mathcal{L}^{\mathrm{Seatbelt}}_{\theta, \phi; \beta, \mathcal{D}}$ using stochastic gradient ascent with minibatches of the data is complicated by the presence of aggregate posteriors $q_\phi(\mathbf{z})$ which depend on the entire dataset. To deal with this, Appendix C we derive a minibatch estimator for TC-penalised hierarchical VAEs, building off that used for $\beta$-TCVAEs (Chen et al., 2018). We note that, as in Chen et al. (2018), large batch sizes are generally required to provide accurate TC estimates.

**Attacking Hierarchical $TC$–Penalised VAEs**   In the above hierarchical model the likelihood over data is conditioned on all layers, so manipulations to any layer have the potential to be significant. We focus on simultaneously attacking all layers, noting that, as shown in Appendix D, this is more effective that just targeting the top or base layers individually. Hence our adversarial objective for latent-space attacks on Seatbelt-VAEs is the following generalisation of that introduced in Tabacof et al. (2016); Gondim-Ribeiro et al. (2018); Kos et al. (2018), to attack all the layers at the same time:

$$\Delta_r^{\mathrm{Seatbelt}}(\mathbf{x}, \mathbf{d}, \mathbf{x}^t; \lambda) = \lambda||\mathbf{d}||_2 + \sum_{i=1}^{L} r(q_\phi(\mathbf{z}^i|\mathbf{x} + \mathbf{d}), q_\phi(\mathbf{z}^i|\mathbf{x}^t)). \quad (10)$$

## 5   EXPERIMENTS

Expanding on the brief experiments in Section 3.2, we perform a battery of adversarial attacks on each of the introduced models. We do this for three different adversarial attacks: first (as in Section 3.2) a latent attack, Eqs (1,10) using the $D_{\mathrm{KL}}$ divergence between attacked and target posteriors; secondly, we attack via the model's output, aiming to make the target maximally likely under the attacked model as in Eq (2); finally, a new latent attack method as per Eqs (1,10) where we use $r(\cdot, \cdot) = W_2(\cdot, \cdot)$, the 2-Wasserstein distance between attacked and target posteriors.

We then evaluate the effectiveness of these attacks in three ways. First, like Fig 1, we can plot the attacks themselves, to see how effective these attacks are in fooling us. Secondly, we can measure the adversary's loss under the attack objective. Thirdly, we give the negative adversarial likelihood of the target image $\mathbf{x}^t$ given an attacked latent representation $\mathbf{z}^*$. Larger, more positive, values of $-\log p_\theta(\mathbf{x}^t|\mathbf{z}^*)$ correspond to less successful attacks as they correspond to large distances between the target and the adversarial reconstruction. Lower values correspond to successful attacks as they correspond to a small distance between the adversarial target and the reconstruction. We also measure

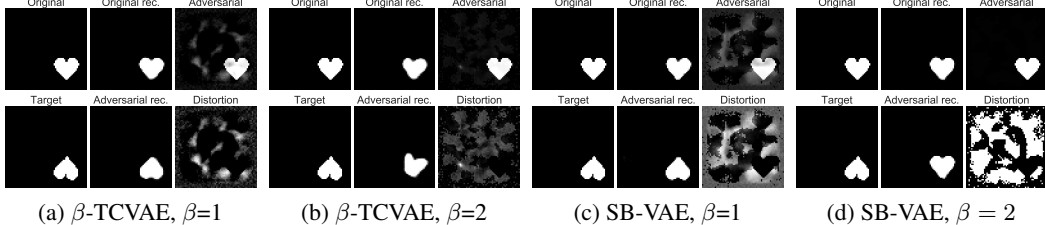

(a) $\beta$-TCVAE, $\beta$=1     (b) $\beta$-TCVAE, $\beta$=2     (c) SB-VAE, $\beta$=1     (d) SB-VAE, $\beta = 2$

Figure 4: $D_{\mathrm{KL}}$ Latent space attacks *only on rotation* of a heart-shaped dSprite for $\beta$-TCVAEs ($d_{\mathbf{z}} = 64$) and Seatbelt-VAEs ($L = 2$) for $\beta = \{1, 2\}$. The attacks are conducted by applying a distortion (third column of each image) to the original image (top first column) to produce an adversarial input (bottom second column of each image) to try to cause the output of the target image (bottom first column). Here we show the most successful adversarial distortion in terms of adversarial loss for each model. It is apparent that Seatbelt-VAEs are the most resilient to attack. Note that the distortions plots (bottom right) are scaled to [0,1] for ease of viewing.

reconstruction quality of these models, as a function of degree of regularisation. Finally, we also measure how downstream tasks that use output of these models perform under attack. We train classifiers, on the reconstructions and on the latent representations, and see how robust performance is when the upstream VAE is attacked.

We demonstrate that hierarchical $TC$–Penalised VAEs (Seatbelt-VAEs) confer superior robustness to $\beta$-TCVAEs and standard VAEs, while preserving the ability to reconstruct inputs effectively. Through this, we demonstrate that they are a powerful tool for learning robust deep generative models.

Following previous work (Tabacof et al., 2016; Gondim-Ribeiro et al., 2018) we randomly sample 10 input-target pairs for each dataset and for each image pair we consider 50 different values of $\lambda$ geometrically-distributed from $2^{-20}$ to $2^{20}$. Thus each individual trained model undergoes 500 attacks for each attack mode. As before, we used L-BFGS-B for gradient descent (Byrd et al., 1995). We perform these experiments on Chairs (Aubry et al., 2014), 3D faces (Paysan et al., 2009), and CelebA (Liu et al., 2015). Details of neural architectures and training are given in Appendix G.

## 5.1 VISUAL APPRAISAL OF ATTACKS

We first visually appraise the effectiveness of attacks that use the $D_{KL}$ divergence on vanilla VAEs, $\beta$-TCVAEs, and Seatbelt-VAEs. As mentioned in Section 1, Fig 1 shows the results of latent space attacks on three models trained on CelebA. It is apparent that the $\beta$-TCVAE provides additional resilience to the attacks compared with the standard VAE. Furthermore, this figure shows that Seatbelt-VAEs are sufficiently robust to almost completely thwart the adversary: its adversarial construction still resembles the original input. Moreover, this was achieved while also producing a clearer non–adversarial reconstruction. One might expect attacks targeting a single generative factor underpinning the data to be easier. However, we find that these models protect effectively against this as well. For example, see Fig 4 for plots showing an attacker attempting to rotate a dSprites heart.

In both figures we follow the method of Gondim-Ribeiro et al. (2018) to plot attacks. Those shown are representative of the adversarial inputs the attacker was able to find over the 50 different values of $\lambda$. The Seatbelt-VAE input only undergoes a small perturbation because it is sufficiently robust that the attacker is not able to make the reconstruction look more like the target image in any meaningful way, such that the optimiser never drifts far from the initial input. Note that the $\beta$-TCVAE is also robust here. The attacker is unable to induce the desired adversarial reconstruction, even though the attack may be of large magnitude. In contrast, attacks on vanilla-VAEs are able to move through the latent space and find a perturbation that reconstructs to the adversary's target image.

## 5.2 QUANTITATIVE ANALYSIS OF ROBUSTNESS

Having ascertained perceptually that Seatbelt-VAEs offer the strongest protection to adversarial attack, we now demonstrate this quantitatively. Fig 5 shows $-\log p_{\theta}(\mathbf{x}^t|\mathbf{z}^*)$ and $\Delta$ over a range of datasets and $\beta$s for Seatbelt-VAEs ($L = 4$) and $\beta$-TCVAEs for our three different attacks. It demonstrates that the combination of depth and high TC-penalisation offers the best protection to

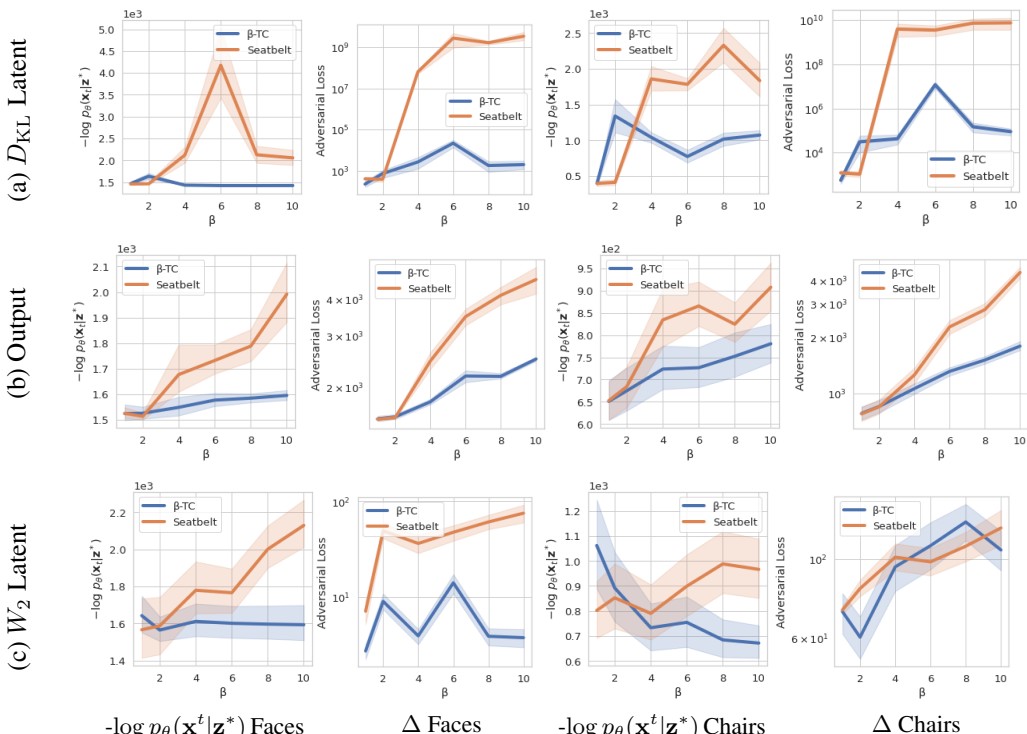

Figure 5: Plots showing the robustness of Seatbelt-VAEs ($L$=4) and $\beta$-TCVAEs models for different values of $\beta$ for three different attack methods: a) Latent space attack via $D_{\mathrm{KL}}$ in Eqs (1,10), b) Attack via the model output as in Eq 2, and c) Latent space attack via the 2-Wasserstein ($W_2$) distance in Eqs (1,10). Note that the $\beta$-TCVAE with $\beta = 1$ corresponds to a vanilla VAE and that $L > 1$ $\beta = 1$ models correspond to hierarchical baselines. We show the negative adversarial likelihood of a target image $\mathbf{x}^t$ given an attacked latent representation $\mathbf{z}^*$ for Faces (1$^{\mathrm{st}}$ col) and Chairs (3$^{\mathrm{rd}}$ col) respectively. Larger values of $-\log p_\theta(\mathbf{x}^t|\mathbf{z}^*)$ mean less successful adversarial attacks. We also show the adversarial loss $\Delta$ in 2$^{\mathrm{nd}}$ and 4$^{\mathrm{th}}$ cols, which have a logarithmic axis. Shading in results corresponds to the 95% CI over variation for 10 images for each combination of $d_{\mathbf{z}} = \{4, 8, 16, 32, 64, 128\}$ and $\lambda$ taking 50 geometrically distributed values between $2^{-20}$ and $2^{20}$.

adversarial attacks and that the hierarchical extension confers much greater protection to adversarial attack than a single layer $\beta$-TCVAE. As we go to the largest values of $\beta$ for both Chairs and 3D Faces, adversarial loss $\Delta_{\mathrm{KL}}$ grows by a factor of $\approx 10^7$ and $-\log p_\theta(\mathbf{x}^t|\mathbf{z}^*)$ for those attacks doubles for Seatbelt-VAE. For all attacks, TC-penalised models outperformed standard VAEs ($\beta$=1) and Seatbelt-VAEs outperform single-layer VAEs. $\beta$-TCVAEs do not experience such a large uptick in adversarial loss and negative adversarial likelihood. These results show that the hierarchical approach can offer very strong protection from the adversarial attacks studied.

In Appendix D we provide plots detailing these metrics for a range of $L$ values. In Appendix E we also calculate the $L_2$ distance between target images and adversarial outputs and show that the loss of effectiveness of adversarial attacks is not due to the degradation of reconstruction quality from increasing $\beta$. We also test VAE robustness to random noise. We noise the inputs and evaluate the model's ability to reconstruct the original input. Through this we are evaluating their ability to denoise. See Appendix F for an illustration of this for TC-penalised models. It is plausible that the ability of these models' to denoise is linked to their robustness to attacks.

**ELBO and Reconstructions** Though Seatbelt-VAEs offer better protection to adversarial attack than $\beta$-TCVAEs, we also motivate their utility by way of their reconstruction quality. In Fig 6 we plot the ELBO of the two TC-penalised models, calculated *without* the $\beta$ penalisation that was applied during training. We further show the effect of depth and TC-penalisation on CelebA reconstructions. These plots show that Seatbelt-VAEs' reconstructions are more resilient to increasing $\beta$ than $\beta$-TCVAEs'.

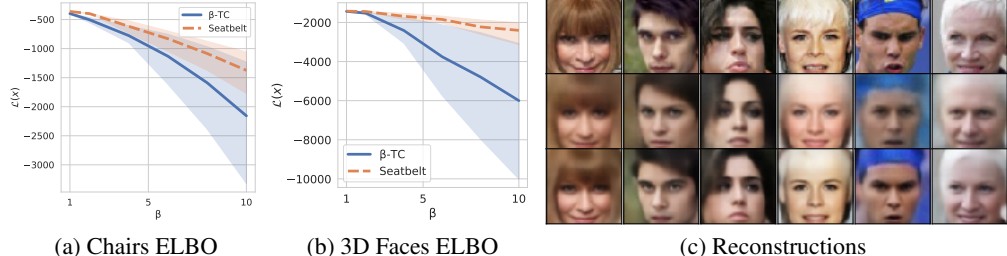

| (a) Chairs ELBO | (b) 3D Faces ELBO | (c) Reconstructions |

Figure 6: Effect of varying $\beta$ on the reconstructions of TC-penalised models. In sub-figures (a) and (b) we plot the final ELBO of TC-penalised models trained on the Chairs and 3D faces, calculated *without* the $\beta$ penalisation applied during training. Shading gives the 95% CI over variation due to variation of $d_{\mathbf{z}} = \{32, 64, 128\}$ for $\beta$-TCVAE and also $L = \{2, 3, 4, 5\}$ for Seatbelt. As $\beta$ increases $\mathcal{L}$ degrades more slowly for Seatbelt-VAE, relative to $\beta$-TCVAE, (c) serves as a visual confirmation of these results. The top row shows CelebA input data. The bottom row, the reconstructions from a Seatbelt-VAE with $L = 4$ and $\beta = 20$, clearly maintains facial identity better than those from a $\beta$-TCVAE, the middle row: many of the individuals' finer facial features lost by the $\beta$-TCVAE are maintained by the Seatbelt-VAE.

Table 1: Robustness of downstream classification tasks under adversarial attack. We consider classifiers trained either on the reconstructed image (denoted $p(y|\tilde{\mathbf{x}})$) or on the latent representations ($p(y|\mathbf{z})$). We show accuracy when the model is attacked, resulting in perturbed embeddings $\mathbf{z}^*$ and reconstructions ($\tilde{\mathbf{x}}^*$). Parentheses show the drop in accuracy resulting from the attack – the smaller the drop in magnitude the better

| Dataset | Task | Accuracy by Model | | |
| --- | --- | --- | --- | --- |
| | | VAE | $\beta$–TCVAE | Seatbelt-VAE |
| SVHN | $p_{\mathrm{MLP}}(y|\tilde{\mathbf{x}})$ | $0.17\,(-0.35)$ | $0.22\,(-0.29)$ | $\mathbf{0.35}\,(\mathbf{-0.15})$ |
| | $p_{\mathrm{Conv}}(y|\tilde{\mathbf{x}})$ | $0.13\,(-0.54)$ | $0.36\,(-0.28)$ | $\mathbf{0.41}\,(\mathbf{-0.26})$ |
| | $p_{\mathrm{MLP}}(y|\mathbf{z})$ | $0.15\,(-0.57)$ | $0.46\,(-0.23)$ | $\mathbf{0.57}\,(\mathbf{-0.21})$ |
| CIFAR10 | $p_{\mathrm{MLP}}(y|\tilde{\mathbf{x}})$ | $0.17\,(-0.32)$ | $0.25\,(-0.21)$ | $\mathbf{0.38}\,(\mathbf{-0.09})$ |
| | $p_{\mathrm{Conv}}(y|\tilde{\mathbf{x}})$ | $0.07\,(-0.37)$ | $0.32\,(-0.10)$ | $\mathbf{0.34}\,(\mathbf{-0.07})$ |
| | $p_{\mathrm{MLP}}(y|\mathbf{z})$ | $0.16\,(-0.41)$ | $0.26\,(-0.23)$ | $\mathbf{0.39}\,(\mathbf{-0.09})$ |

## 5.3 Protection to Downstream Tasks

Finally, we consider the protection that Seatbelt-VAEs might provide to downstream tasks, noting that VAEs are often used as subcomponents in larger ML systems (Higgins et al., 2017b), or as a mechanism to protect another model from attack (Schott et al., 2019; Ghosh et al., 2019). Table 1 shows results for classification tasks using 2-layer MLPs and fully-convolutional nets trained on the reconstructions or on the embeddings. It shows the drop in accuracy caused by an adversary that picks a target with a different label and attacks the VAEs' embedding using the attack objective with $\lambda = 1$. We see that Seatbelt-VAEs produced significantly better accuracies under these attacks.

## 6 Conclusion

We have shown that VAEs can be rendered more robust to adversarial attacks by regularising the evidence lower bound. This increase in robustness can be strengthened by extending these regularisation methods to hierarchical VAEs, forming Seatbelt-VAEs, which uses a generative structure where the likelihood makes use of all the latent variables. Designing robust VAEs is becoming pressing as they are increasingly deployed as subcomponents in larger pipelines. As we have shown, methods typically used for disentangling, motivated by their ability to provide interpretable representations, also confer robustness. Studying the beneficial effects of these methods is starting to come to the fore of VAE research.

ACKNOWLEDGEMENTS

This research was directly funded by the Alan Turing Institute under Engineering and Physical Sciences Research Council (EPSRC) grant EP/N510129/1. MW was supported by EPSRC grant EP/G03706X/1. AC was supported by an EPSRC Studentship. SR gratefully acknowledges support from the UK Royal Academy of Engineering and the Oxford-Man Institute. CH was supported by the Medical Research Council, the Engineering and Physical Sciences Research Council, Health Data Research UK, and the Li Ka Shing Foundation

We thank Tomas Lazauskas, Jim Madge and Oscar Giles from the Alan Turing Institute's Research Engineering team for their help and support.

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

# A   VARIATIONAL AUTOENCODERS

Variational autoencoders (VAEs) are a variety of generative model suitable for high-dimensional data like images (Kingma & Welling, 2014; Rezende et al., 2014). They introduce a joint distribution over data $\mathbf{x}$ and latent variables $\mathbf{z}$: $p_\theta(\mathbf{x}, \mathbf{z}) = p_\theta(\mathbf{x}|\mathbf{z})p(\mathbf{z})$, where $p_\theta(\mathbf{x}|\mathbf{z})$ is an appropriate distribution given the form of the data, the parameters of which are represented by deep nets with parameters $\theta$, and $p(\mathbf{z}) = \mathcal{N}(0, \mathcal{I})$ is a common choice for the prior. As exact inference is intractable, one performs amortised stochastic variational inference by introducing an inference network for the latent variables, $q_\phi(\mathbf{z}|\mathbf{x})$, which often also takes the form of a Gaussian, $\mathcal{N}(\mathbf{z}|\mu_\phi(\mathbf{x}), \Sigma_\phi(\mathbf{x}))$. We can then perform gradient ascent, with respect to both $\theta$ and $\phi$, on the evidence lower bound (ELBO)

$$\mathcal{L}(\mathbf{x}) = \mathbb{E}_{q_\phi(\mathbf{z}|\mathbf{x})} \left[ \log p_\theta(\mathbf{x}|\mathbf{z}) \right] - D_{\mathrm{KL}}(q_\phi(\mathbf{z}|\mathbf{x})||p(\mathbf{z})), \tag{11}$$

using the reparameterisation trick to take gradients through Monte Carlo samples from $q_\phi(\mathbf{z}|\mathbf{x})$.

## A.1   DISENTANGLING VAEs

When learning *disentangled* representations (Bengio et al., 2013) in a VAE, one attempts to establish a one-to-one correspondence between dimensions of the learnt latent space and some interpretable aspect of the data (Higgins et al., 2017a; Burgess et al., 2017; Chen et al., 2018; Mathieu et al., 2019). One dimension of the latent space could encode the rotation of a face for instance. Mathieu et al. (2019) offers a broader perspective, where disentangling can be interpreted as a particular case of *decomposition*. In decomposition, models have the right degree of overlap between their latent posteriors such that the aggregate posterior matches the prior well throughout the latent space $\mathcal{Z}$.

Disentangling is often enforced by an added penalisation to the VAE ELBO that acts akin to a regularisation method. Because of this, disentangling can be difficult to achieve in practice, and often requires precisely choosing the hyperparameters of the model *and* of the weighting of the added regularisation term (Locatello et al., 2019; Mathieu et al., 2019; Rolinek et al., 2019). That disentangling relies on forms of soft supervision renders the task of learning disentangled representations potentially problematic (Khemakhem et al., 2020). When viewed as a purely unsupervised task it can be hard to establish a direct correspondence between a disentangling-VAE's training objective and the learning of a disentangled latent space. Nevertheless, models trained under disentangling objectives have other beneficial properties. For example, the encoders of some disentangled VAEs have been used as the perceptual part of deep reinforcement learning models to create agents more robust to variation in their environment (Higgins et al., 2017b). Thus, regardless of the presence of disentangled generative factors, these regularisation methods can be useful for downstream tasks. In this paper we show that methods developed to obtain disentangled representations have the benefit of conferring robustness to adversarial attack.

A commonly used disentangling method is that of the $\beta$-VAE. In a $\beta$-VAE (Higgins et al., 2017a), a free parameter $\beta$ multiplies the $D_{\mathrm{KL}}$ term in the evidence lower bound $\mathcal{L}(\mathbf{x})$. This objective $\mathcal{L}_\beta(\mathbf{x})$ remains a lower bound on the evidence:

$$\mathcal{L}_\beta(\mathbf{x}) := \mathbb{E}_{q_\phi(\mathbf{z}|\mathbf{x})} \left[ \log p_\theta(\mathbf{x}|\mathbf{z}) \right] - \beta D_{\mathrm{KL}}(q_\phi(\mathbf{z}|\mathbf{x})||p(\mathbf{z})) \right]$$

The $\beta$-VAE though it offers a simple method for obtaining potentially disentangled representations does so at the expense of model quality. Models trained with large $\beta$ penalisation suffer from poor quality reconstructions and lower ELBO. For more discussion of their theoretical aspects, see Kumar & Poole (2020).

Other methods seek to offset this degradation in model quality by decomposing the ELBO and more precisely targeting the regularisation when obtaining disentangled representations. We can more insight into VAEs by defining the evidence lower bound not per data-point, but instead over the dataset $\mathcal{D}$ of size $N$, $\mathcal{D} = \{\mathbf{x}^n\}$, so we have $\mathcal{L}(\theta, \phi, \mathcal{D})$ (Hoffman & Johnson, 2016; Makhzani et al., 2016; Kim & Mnih, 2018; Chen et al., 2018; Esmaeili et al., 2019). From this, Esmaeili et al. (2019)

gives a decomposition of the dataset-level evidence lower bound:

$$\mathcal{L}(\theta, \phi, \mathcal{D}) = \mathbb{E}_{q_\phi(\mathbf{z}, \mathbf{x})} \log \frac{p_\theta(\mathbf{x}, \mathbf{z})}{q_\phi(\mathbf{z}, \mathbf{x})} \tag{12}$$

$$= \mathbb{E}_{q_\phi(\mathbf{z}, \mathbf{x})} \Big[ \underbrace{\log \frac{p_\theta(\mathbf{x}|\mathbf{z})}{p_\theta(\mathbf{x})}}_{①} - \underbrace{\log \frac{q_\phi(\mathbf{z}|\mathbf{x})}{q_\phi(\mathbf{z})}}_{②} \Big] - \underbrace{D_{\mathrm{KL}}(q(\mathbf{x})||p_\theta(\mathbf{x}))}_{③} - \underbrace{D_{\mathrm{KL}}(q_\phi(\mathbf{z})||p(\mathbf{z}))}_{④} \tag{13}$$

where under the assumption that $p(\mathbf{z})$ factorises we can further decompose ④:

$$D_{\mathrm{KL}}(q_\phi(\mathbf{z})||p(\mathbf{z})) = \mathbb{E}_{q_\phi(\mathbf{z})} \Big[ \underbrace{\log \frac{q_\phi(\mathbf{z})}{\prod_j q_\phi(\mathbf{z}_j)}}_{Ⓐ} \Big] + \sum_j \underbrace{D_{\mathrm{KL}}(q_\phi(\mathbf{z}_j)||p(\mathbf{z}_j))}_{Ⓑ} \tag{14}$$

where $j$ indexes over coordinates in $\mathbf{z}$. $q_\phi(\mathbf{z}, \mathbf{x}) = q_\phi(\mathbf{z}|\mathbf{x})q(\mathbf{x})$ and $q(\mathbf{x}) := \frac{1}{N}\sum_{n=1}^{N} \delta(\mathbf{x} - \mathbf{x}^n)$ is the empirical data distribution. $q_\phi(\mathbf{z}) := \frac{1}{N}\sum_{n=1}^{N} q_\phi(\mathbf{z}|\mathbf{x}^n)$ is called the aggregate posterior.

Ⓐ is the total correlation (TC) for $q_\phi(\mathbf{z})$.

**Definition A.1.** *The total correlation (TC) is a generalisation of mutual information to multiple variables (Watanabe, 1960) and is often used as the objective Independent Component Analysis (Bell & Sejnowski, 1995). The TC is defined as is defined as the KL divergence from the joint distribution $p(\mathbf{s}), \mathbf{s} \in \mathbb{R}^d$ to the independent distribution over the dimensions of the variable $\mathbf{s}$: $p(\mathbf{s}_1)p(\mathbf{s}_2)\ldots p(\mathbf{s}_n)$. Formally: $\mathrm{TC}(\mathbf{s}) = D_{\mathrm{KL}}(p(\mathbf{s})|| \prod_{j=1}^{d} p(\mathbf{s}_j))$*

With this mean-field $p(\mathbf{z})$, Factor and $\beta$-TCVAEs upweight the TC of the aggregate posterior, so we have an objective:

$$\mathcal{L}^{\beta\mathrm{TC}}(\theta, \phi, \mathcal{D}) = ① + ② + ③ + Ⓑ + \beta Ⓐ \tag{15}$$

Upweighting the penalisation associated with the TC term promotes the learning of independent latent factors, one of the key objectives of disentangling. Chen et al. (2018) show empirically that the learnt representations are disentangled when the hyperparameters of the model are well-chosen. They also give a differentiable, stochastic approximation to $\mathbb{E}_{q_\phi(\mathbf{z})} \log q_\phi(\mathbf{z})$, rendering this decomposition simple to use as a training objective using stochastic gradient descent. However this is a biased estimator: it is a nested expectation, for which unbiased, finite–variance, estimators do not generally exist (Rainforth et al., 2018). Consequently, it has the unfortunate consequence of needing large batch sizes to have the desired behaviour; for small batch sizes its practical behaviour mimics that of the $\beta$-VAE (Mathieu et al., 2019).

## A.2    $\beta$-VAEs, $TC$-Penalisation and Overlap

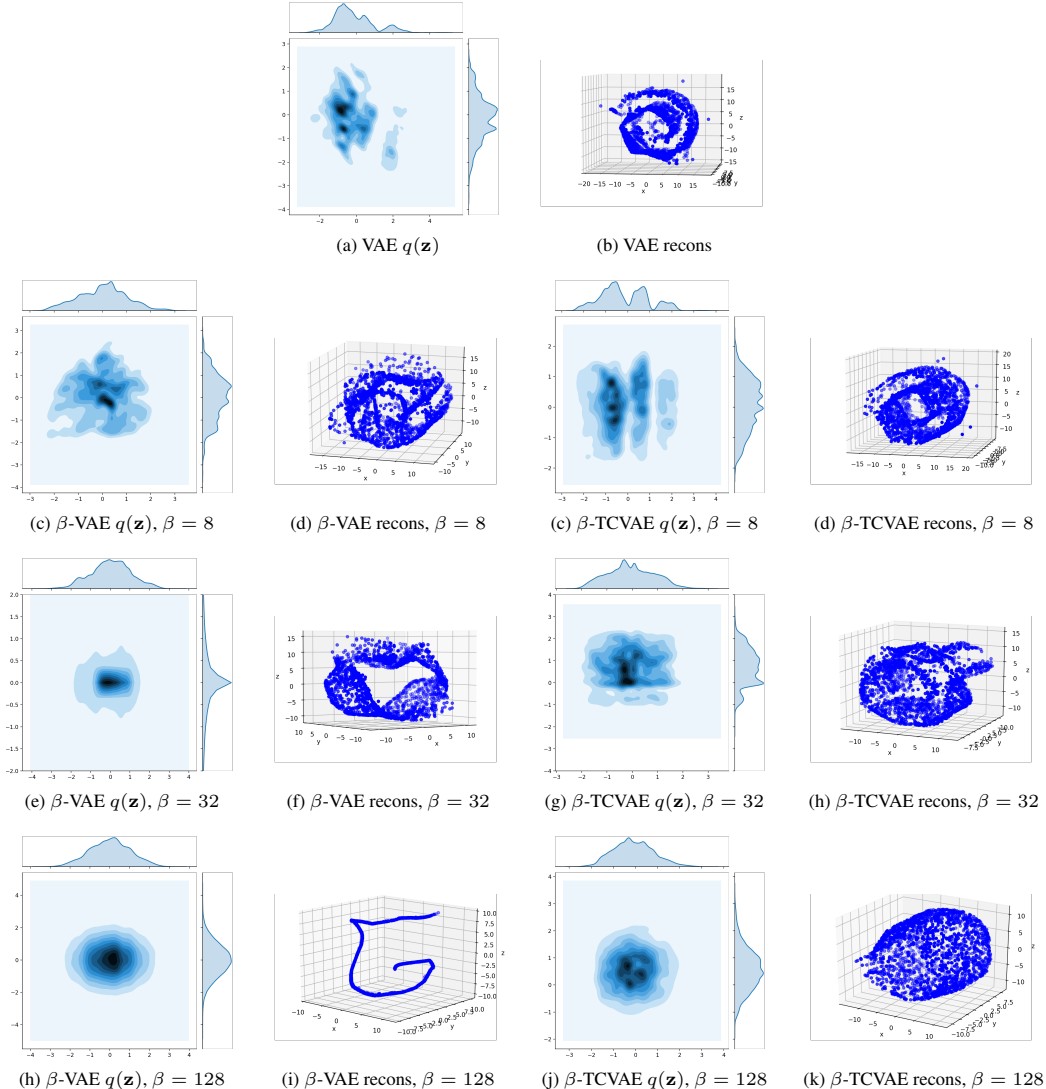

Figure A.7: $\beta$-VAEs and $\beta$-TCVAEs trained on 3D 'Swiss Roll' data, with a vanilla VAE as baseline and all with 2D latents. $\beta \in \{8, 32, 128\}$. The aggregate posteriors, for both model types, tend to become smoother as $\beta$ increases. Note, however, that for large $\beta$ values the $\beta$-VAEs suffer a catastrophic collapse in performance (in terms of reconstructions), while the $\beta$-TCVAEs degrade more gracefully. The requirement that $\beta$-TCVAEs upweight, that the aggregate posterior is well-approximated by the produce of its dimensionwise marginals, is clearly much less onerous to achieve while still modelling the data well than that of $\beta$-VAEs, which requires each datapoint's amortised posterior to closely match the prior.

Recall from the discussion in § 3 that it is gaps, holes, in the aggregate posterior that adversaries can exploit. We also want to close up these holes without degrading the model too much. Rezende & Viola (2018) observed that in regions of $\mathcal{Z}$ when the aggregate posterior places no density the decoder is unconstrained by the ELBO. It is these regions, with associated unconstrained decoder behaviour, that enable adversaries to have an easy time attacking the model. Thus our aim in making robust VAEs is to have an aggregate posterior that is smooth in the sense of having relatively flat density across $\mathcal{Z}$, so therefore having no holes. This is equivalent to overlap, as introduced in Mathieu et al. (2019).

So, why do these regularisation methods increase overlap? Why can upweighting penalisation of the Total Correlation – demanding that the aggregate posterior is well-approximated by the product of its marginals – be expected to increase overlap? And why it does so in a superior way to a $\beta-$VAE's upweighting of $D_{\mathrm{KL}}(q_\phi(\mathbf{z}|\mathbf{x})||p(\mathbf{z}))$? Recall that in Fig 2 we showed that the $L_2$ norm of the standard deviation of the encoder concentrates at a particular value for $\beta$-VAEs, but for $\beta$-TCVAEs it takes a broader ranger of values, values above the saturation point of $\beta$-VAEs.

In a $\beta$-VAE with large $\beta$ we are asking that the amortised posterior is close to the prior for all inputs. So for $p(\mathbf{z}) = \mathcal{N}(0,1)$ we are forcing $\boldsymbol{\mu}_\phi(\mathbf{x})$ to 0 and $\boldsymbol{\sigma}_\phi(\mathbf{x})$ to 1. Naturally this will lead our aggregate posterior to have a high degree of overlap between its constituent mixture components, because all of them are being driven to be the same. And with all per-datapoint posteriors being driven to be the same, information about the initial input data is necessarily lost in these representations.

For a $\beta$-TCVAE, however, the demand for the aggregate posterior to be well-approximated by the product of its marginals does not in itself entail a fixed scale, nor does it push all the per-datapoint posteriors towards the prior. Rather we are directly asking for statistical independence between coordinate directions. Holes in the aggregate posterior are (as long as they are off-axis) a form of dependency between the latent variables. By demanding that the aggregate posterior factorises, we are thus asking the model to 'smooth out' any holes (or peaks) that do not lie along the axes of the latent space. Intuitively, and as shown in Figure 2, can be done achieved without causing as strong degradation to model quality, as measured by the fidelity of reconstructions and the values of the ($\beta = 1$) ELBO.

To give us a more direct understanding here we perform some toy experiments on 'Swiss Roll' data, Fig A.7. We train 2D-latent-space VAEs: vanilla, $\beta$-VAEs, and $\beta$-TCVAEs. We plot the aggregate posterior and the reconstructions (the means of the likelihood conditioned on a sample of each per-datapoint posterior). Clearly the amount of overlap increases with $\beta$ for both kinds of model, but the $\beta$-TCVAEs seem to do this in a more structured way and, unlike the $\beta$-VAE, does not suffer from (eventually catastrophic) degradation in model quality for large $\beta$.

## A.3 HIERARCHICAL VAES

In a hierarchical VAE we have a set of $L$ layers of $\mathbf{z}$ variables: $\vec{\mathbf{z}} = \{\mathbf{z}^i\}$. However, training DLGMs is challenging: the latent variables furthest from the data can fail to learn anything informative (Sønderby et al., 2016; Zhao et al., 2017). Due to the factorisation of $q_\phi(\vec{\mathbf{z}}|\mathbf{x})$ and $p_\theta(\mathbf{x}, \vec{\mathbf{z}})$ in a DLGM, it is possible for a single-layer VAE to train in isolation within a hierarchical model: each $p_\theta(\mathbf{z}^i|\mathbf{z}^{i+1})$ distribution can become a fixed distribution not depending on $\mathbf{z}^{i+1}$ such that each $D_{\mathrm{KL}}$ divergence present in the objective between corresponding $\mathbf{z}^i$ layers can still be driven to a local minima. (Zhao et al., 2017) gives a proof of this separation for the case where the model is perfectly trained, i.e. $D_{\mathrm{KL}}(q_\phi(\mathbf{z}, \mathbf{x})||p_\theta(\mathbf{x}, \mathbf{z})) = 0$.

This is the hierarchical version of the collapse of $\mathbf{z}$ units in a single-layer VAE (Burda et al., 2016), but now the collapse is over entire layers $\mathbf{z}^i$. It was part of the motivation for the Ladder VAE (Sønderby et al., 2016) and BIVA (Maaløe et al., 2019).

More recently Vahdat & Kautz (2020); Child (2021) have shown that by judicious neural parameterisation and training strategy, hierarchical VAEs can obtain $\sim$SOTA results in the probabilistic modelling and generation of images.

# B  TOTAL-CORRELATION DECOMPOSITION OF ELBO

Proof of Theorem 1

Here we prove that the ELBO for a hierarchical VAE with forward model as in Eq (3) and amortised variational posterior as in Eq (4) can be decomposed to reveal a total-correlation in the top-most latent variable.

Specifically, now considering the ELBO for the whole dataset and using $q(\mathbf{x})$ to indicate the empirical data distribution, we will obtain, denoting $\mathbf{z}^0 = \mathbf{x}$:

$$
\begin{aligned}
\mathcal{L}(\theta, \phi; \mathcal{D}) = {} & \mathbb{E}_{q_\phi(\vec{\mathbf{z}}, \mathbf{x})}\left[\log p_\theta(\mathbf{x}|\vec{\mathbf{z}})\right] - \mathbb{E}_{q_\phi(\vec{\mathbf{z}}|\mathbf{x})q(\mathbf{x})}\left[\sum_{i=1}^{L-1} D_{\mathrm{KL}}(q_\phi(\mathbf{z}^i|\mathbf{z}^{i-1}, \mathbf{x})||p_\theta(\mathbf{z}^i|\mathbf{z}^{i+1}))\right] \\
& - \mathbb{E}_{q_\phi(\mathbf{z}^{L-1})} D_{\mathrm{KL}}(q_\phi(\mathbf{z}^L, \mathbf{x}|\mathbf{z}^{L-1})||q_\phi(\mathbf{z}^L)q(\mathbf{x})) \\
& - \sum_j D_{\mathrm{KL}}(q_\phi(\mathbf{z}_j^L)||p(\mathbf{z}_j^L)) - \beta D_{\mathrm{KL}}\left(q_\phi(\mathbf{z}^L)||\prod_j q_\phi(\mathbf{z}_j^L)\right)
\end{aligned}
\tag{16}
$$

We start with the forms of $p$ and $q$ given in Theorem 1. The likelihood is conditioned on all $\mathbf{z}$ layers: $p_\theta(\mathbf{x}|\vec{\mathbf{z}})$.

$$
\mathcal{L}(\theta, \phi; \mathcal{D}) = \mathbb{E}_{q_\phi(\vec{\mathbf{z}}, \mathbf{x})} \log \frac{p_\theta(\mathbf{x}, \vec{\mathbf{z}})}{q_\phi(\vec{\mathbf{z}}, \mathbf{x})}
\tag{17}
$$

$$
= \mathbb{E}_{q_\phi(\vec{\mathbf{z}}, \mathbf{x})}\left[\log p_\theta(\mathbf{x}|\vec{\mathbf{z}})\right] - \mathbb{E}_{q(\mathbf{x})}\left[D_{\mathrm{KL}}(q_\phi(\vec{\mathbf{z}}, \mathbf{x})||p_\theta(\vec{\mathbf{z}}))\right]
\tag{18}
$$

$$
= \mathbb{E}_{q(\vec{\mathbf{z}}, \mathbf{x})} \log p_\theta(\mathbf{x}|\vec{\mathbf{z}}) - \mathbb{E}_{q(\mathbf{x})} \log q(\mathbf{x}) + \mathbb{E}_{q(\vec{\mathbf{z}}, \mathbf{x})} \log \frac{p_\theta(\vec{\mathbf{z}})}{q(\vec{\mathbf{z}}|\mathbf{x})}
\tag{19}
$$

$$
= \mathbb{E}_{q(\vec{\mathbf{z}}, \mathbf{x})} \log p_\theta(\mathbf{x}|\vec{\mathbf{z}}) + \mathcal{H}(q(\mathbf{x}))
\tag{20}
$$

$$
+ \underbrace{\int d\mathbf{x}\, d\mathbf{z}^1 \prod_{i=2}^{L} (d\mathbf{z}^i q_\phi(\mathbf{z}^i|\mathbf{z}^{i-1}, \mathbf{x})) q_\phi(\mathbf{z}^1|\mathbf{x}) q(\mathbf{x}) \log \frac{p(\mathbf{z}^L)\prod_{k=1}^{L-1} p_\theta(\mathbf{z}^k|\mathbf{z}^{k+1})}{q_\phi(\mathbf{z}^1|\mathbf{x})\prod_{m=1}^{L-1} q_\phi(\mathbf{z}^{m+1}|\mathbf{z}^m, \mathbf{x})}}_{\text{\textcircled{W}}}
$$

So here we have three terms: an expectation over the data likelihood, the entropy of the empirical data distribution (a constant) and \textcircled{W}. We now can expand \textcircled{W} to a term involving the prior for the latent $\mathbf{z}^L$ and a term involving the conditional distributions from the generative model for the remaining components of $\vec{\mathbf{z}}$:

$$
\textcircled{W} = \underbrace{\int d\mathbf{x} \prod_{i=1}^{L} (d\mathbf{z}^i) q_\phi(\vec{\mathbf{z}}|\mathbf{x}) q(\mathbf{x}) \log \frac{\prod_{k=1}^{L-1} p_\theta(\mathbf{z}^k|\mathbf{z}^{k+1})}{q_\phi(\mathbf{z}^1|\mathbf{x})\prod_{m=1}^{L-2} q_\phi(\mathbf{z}^{m+1}|\mathbf{z}^m, \mathbf{x})}}_{\text{\textcircled{R}}}
$$

$$
+ \underbrace{\int d\mathbf{x} \prod_{i=1}^{L} (d\mathbf{z}^i) q_\phi(\vec{\mathbf{z}}|\mathbf{x}) q(\mathbf{x}) \log \frac{p(\mathbf{z}^L)}{q_\phi(\mathbf{z}^L|\mathbf{z}^{L-1}, \mathbf{x})}}_{\text{\textcircled{S}}}
\tag{21}
$$

The first part \textcircled{R}, it that part of \textcircled{W} not involving the prior for 'top-most' latent variable $\mathbf{z}^L$, is the first subject of our attention. We split out the part of \textcircled{R} involving the generative and posterior terms for the latent variable closest to the data, $\mathbf{z}^1$ and the rest:

$$
\textcircled{R} = \underbrace{\int d\mathbf{x} \prod_{i=1}^{L} (d\mathbf{z}^i) q_\phi(\vec{\mathbf{z}}|\mathbf{x}) q(\mathbf{x}) \log \frac{p_\theta(\mathbf{z}^1|\mathbf{z}^2)}{q_\phi(\mathbf{z}^1|\mathbf{x})}}_{\text{\textcircled{$R_a$}}} + \underbrace{\sum_{m=2}^{L-1} \int d\mathbf{x} \prod_{i=1}^{L} (d\mathbf{z}^i) q_\phi(\vec{\mathbf{z}}|\mathbf{x}) q(\mathbf{x}) \log \frac{p_\theta(\mathbf{z}^m|\mathbf{z}^{m+1})}{q_\phi(\mathbf{z}^m|\mathbf{z}^{m-1}, \mathbf{x})}}_{\text{\textcircled{$R_b$}}}.
$$

The first of these terms \textcircled{$R_a$} is an expectation over a $D_{\mathrm{KL}}$:

$$
\textcircled{$R_a$} = - \mathbb{E}_{q_\phi(\mathbf{z}^2, \mathbf{x})} D_{\mathrm{KL}}(q_\phi(\mathbf{z}^1|\mathbf{x})||p_\theta(\mathbf{z}^1|\mathbf{z}^2)).
\tag{22}
$$

And the rest, $\boxed{R_b}$, provides the $D_{\mathrm{KL}}$ divergences in the ELBO for all latent variables other than $\mathbf{z}^L$ and $\mathbf{z}^1$. It reduces to a sum of expectations over $D_{\mathrm{KL}}$ divergences, one per latent variable.

$$\boxed{R_b} = \sum_{m=2}^{L-1} \int \mathrm{d}\mathbf{x} \prod_{i=1}^{L}(\mathrm{d}\mathbf{z}^i) q_\phi(\mathbf{z}^1|\mathbf{x}) q(\mathbf{x}) \prod_{k=1,\neq m}^{L-1}(q_\phi(\mathbf{z}^{k+1}|\mathbf{z}^k,\mathbf{x})) q_\phi(\mathbf{z}^m|\mathbf{z}^{m-1},\mathbf{x}) \log \frac{p_\theta(\mathbf{z}^m|\mathbf{z}^{m+1})}{q_\phi(\mathbf{z}^m|\mathbf{z}^{m-1},\mathbf{x})} \tag{23}$$

$$= -\sum_{m=2}^{L-1} \int \mathrm{d}\mathbf{x} \prod_{i=1}^{L}(\mathrm{d}\mathbf{z}^i) q_\phi(\mathbf{z}^1|\mathbf{x}) q(\mathbf{x}) \prod_{k=1,\neq m}^{L-1}(q_\phi(\mathbf{z}^{k+1}|\mathbf{z}^k,\mathbf{x})) D_{\mathrm{KL}}(q_\phi(\mathbf{z}^m|\mathbf{z}^{m-1},\mathbf{x})||p_\theta(\mathbf{z}^m|\mathbf{z}^{m+1})) \tag{24}$$

$$= -\sum_{m=2}^{L-1} \mathbb{E}_{q_\phi(\mathbf{z}^{m+1},\mathbf{z}^{m-1},\mathbf{x})} D_{\mathrm{KL}}(q_\phi(\mathbf{z}^m|\mathbf{z}^{m-1},\mathbf{x})||p_\theta(\mathbf{z}^m|\mathbf{z}^{m+1})). \tag{25}$$

Now we have:

$$\mathcal{L}(\theta,\phi;\mathcal{D}) = \mathbb{E}_{q(\vec{\mathbf{z}},\mathbf{x})} \log p_\theta(\mathbf{x}|\vec{\mathbf{z}}) + \mathcal{H}(q(\mathbf{x})) + \boxed{R_a} + \boxed{R_b} + \boxed{S} \tag{26}$$

We wish to apply TC decomposition to the top-most latent variable $\mathbf{z}^L$. $\boxed{S}$ is an expectation over the $D_{\mathrm{KL}}$ divergence between $q_\phi(\mathbf{z}^L|\mathbf{z}^{L-1},\mathbf{x})$ and $p(\mathbf{z}^L)$

$$\boxed{S} = -\mathbb{E}_{q_\phi(\mathbf{z}^{L-1},\mathbf{x})} D_{\mathrm{KL}}(q_\phi(\mathbf{z}^L|\mathbf{z}^{L-1},\mathbf{x})||p(\mathbf{z}^L)) \tag{27}$$

Applying the decomposition, with $j$ indexes over units in $\mathbf{z}^L$.

$$\boxed{S} = -\mathbb{E}_{q_\phi(\mathbf{z}^L,\mathbf{z}^{L-1},\mathbf{x})} \big[ \log q_\phi(\mathbf{z}^L|\mathbf{z}^{L-1},\mathbf{x}) - \log p(\mathbf{z}^L) + \log q_\phi(\mathbf{z}^L)$$
$$- \log q_\phi(\mathbf{z}^L) + \log \prod_j q_\phi(\mathbf{z}_j^L) - \log \prod_j q_\phi(\mathbf{z}_j^L)\big]$$

$$= -\mathbb{E}_{q_\phi(\mathbf{z}^L,\mathbf{z}^{L-1},\mathbf{x})} \left[\log \frac{q_\phi(\mathbf{z}^L|\mathbf{z}^{L-1},\mathbf{x})}{q_\phi(\mathbf{z}^L)}\right] - \mathbb{E}_{q_\phi(\mathbf{z}^L)} \left[\log \frac{q_\phi(\mathbf{z}^L)}{\prod_j q_\phi(\mathbf{z}_j^L)}\right]$$

$$- \mathbb{E}_{q_\phi(\mathbf{z}^L)} \left[\log \frac{\prod_j q_\phi(\mathbf{z}_j^L)}{p(\mathbf{z}^L)}\right]$$

$$= -\mathbb{E}_{q_\phi(\mathbf{z}^L,\mathbf{z}^{L-1},\mathbf{x})} \left[\log \frac{q_\phi(\mathbf{z}^L|\mathbf{z}^{L-1},\mathbf{x})q(\mathbf{x})}{q_\phi(\mathbf{z}^L)q(\mathbf{x})}\right] - \mathbb{E}_{q_\phi(\mathbf{z}^L)} \left[\log \frac{q_\phi(\mathbf{z}^L)}{\prod_j q_\phi(\mathbf{z}_j^L)}\right]$$

$$- \sum_j \mathbb{E}_{q_\phi(\mathbf{z}^L)} \left[\log \frac{q_\phi(\mathbf{z}_j^L)}{p(\mathbf{z}_j^L)}\right]$$

$$= \underbrace{-\mathbb{E}_{q_\phi(\mathbf{z}^{L-1})}) D_{\mathrm{KL}}(q_\phi(\mathbf{z}^L,\mathbf{x}|\mathbf{z}^{L-1})||q_\phi(\mathbf{z}^L)q(\mathbf{x}))}_{\boxed{S_a}}$$

$$\underbrace{-\sum_j D_{\mathrm{KL}}(q_\phi(\mathbf{z}_j^L)||p(\mathbf{z}_j^L))}_{\boxed{S_b}} \underbrace{-D_{\mathrm{KL}}(q_\phi(\mathbf{z}^L)||\prod_j q_\phi(\mathbf{z}_j^L))}_{\boxed{S_c}}$$

Where we have used $p(\mathbf{z}^L) = \prod_j p(\mathbf{z}_j^L)$ for our chosen generative model, a product of independent unit-variance Gaussian distributions.

$$\mathcal{L}(\theta,\phi;\mathcal{D}) = \mathbb{E}_{q(\vec{\mathbf{z}},\mathbf{x})} \log p_\theta(\mathbf{x}|\vec{\mathbf{z}}) + \mathcal{H}(q(\mathbf{x})) + \boxed{R_a} + \boxed{R_b} + \boxed{S_a} + \boxed{S_b} + \boxed{S_c} \tag{28}$$

Giving us a decomposition of the evidence lower bound that reveals the TC-term in $\mathbf{z}^L$, as required. Multiplying this with a chosen pre-factor $\beta$ gives us the required form. $\square$

## C  MINIBATCH WEIGHTED SAMPLING

As in Chen et al. (2018), applying $\beta$-TC decomposition requires us to calculate terms of the form:

$$\mathbb{E}_{q_\phi(\mathbf{z}^i)} \log q_\phi(\mathbf{z}^i) \tag{29}$$

The $i = 1$ case is covered in the appendix of Chen et al. (2018). First we will repeat the argument for $i = 1$ as made in Chen et al. (2018), but in our notation, and then we cover the case $i > 1$ for models with factorisation of $q_\phi(\vec{\mathbf{z}}|\mathbf{x})$ of Seatbelt VAEs.

### C.1  MWS FOR $\beta$-TCVAEs

We denote $\mathcal{B}_M = \{\mathbf{x}_1, \mathbf{x}_2, ..., \mathbf{x}_M\}$, a minibatch of datapoints drawn uniformly iid from $q(\mathbf{x}) = 1/N \sum_{n=1}^N \delta(\mathbf{x} - \mathbf{x}_n)$. For any minibatch we have $p(\mathcal{B}_M) = \frac{1}{N}^M$. Chen et al. (2018) introduce $r(\mathcal{B}_M|\mathbf{x})$, the probability of a sampled minibatch given that one member is $x$ and the remaining $M - 1$ points are sampled iid from $q(\mathbf{x})$, so $r(\mathcal{B}_M|\mathbf{x}) = \frac{1}{N}^{M-1}$.

$$\mathbb{E}_{q_\phi(\mathbf{z}^1)} \log q_\phi(\mathbf{z}^1) = \mathbb{E}_{q_\phi(\mathbf{z}^1, \mathbf{x})} [\log \mathbb{E}_{q(\mathbf{x})} [q_\phi(\mathbf{z}^1|\mathbf{x})]] \tag{30}$$

$$= \mathbb{E}_{q_\phi(\mathbf{z}^1, \mathbf{x})} [\log \mathbb{E}_{p(\mathcal{B}_M)} [\frac{1}{M} \sum_{m=1}^M q_\phi(\mathbf{z}^1|\mathbf{x}_m)]] \tag{31}$$

$$\geq \mathbb{E}_{q_\phi(\mathbf{z}^1, \mathbf{x})} [\log \mathbb{E}_{r(\mathcal{B}_M|\mathbf{x})} [\frac{p(\mathcal{B}_M)}{r(\mathcal{B}_M|\mathbf{x})} \frac{1}{M} \sum_{m=1}^M q_\phi(\mathbf{z}^1|\mathbf{x}_m)]] \tag{32}$$

$$= \mathbb{E}_{q_\phi(\mathbf{z}^1, \mathbf{x})} [\log \mathbb{E}_{r(\mathcal{B}_M|\mathbf{x})} [\frac{1}{NM} \sum_{m=1}^M q_\phi(\mathbf{z}^1|\mathbf{x}_m)]] \tag{33}$$

$$\tag{34}$$

So then during training, one samples a minibatch $\{\mathbf{x}_1, \mathbf{x}_2, ..., \mathbf{x}_M\}$ and can estimate $\mathbb{E}_{q_\phi(\mathbf{z}^1)} \log q_\phi(\mathbf{z}^1)$ as:

$$\mathbb{E}_{q_\phi(\mathbf{z}^1)} \log q_\phi(\mathbf{z}^1) \approx \frac{1}{M} \sum_{i=1}^M [\log \sum_{j=1}^M q_\phi(\mathbf{z}_i^1|\mathbf{x}_j) - \log NM] \tag{35}$$

and $\mathbf{z}_i^1$ is a sample from $q_\phi(\mathbf{z}^1|\mathbf{x}_i)$.

### C.2  MINIBATCH WEIGHTED SAMPLING FOR SEATBELT-VAEs

Here we have that $q(\vec{\mathbf{z}}, \mathbf{x}) = \prod_{l=2}^L [q_\phi(\mathbf{z}^l|\mathbf{z}^{l-1}, \mathbf{x})] q_\phi(\mathbf{z}^1|\mathbf{x}) q(\mathbf{x})$. Now instead of having a minibatch of datapoints, we have a minibatch of draws of $\mathbf{z}^{i-1}$: $\mathcal{B}_M^{i-1} = \{\mathbf{z}_1^{i-1}, \mathbf{z}_2^{i-1}, ..., \mathbf{z}_M^{i-1}\}$. Each member of which is the result of sequentially sampling along a chain, starting with some particular datapoint $\mathbf{x}_m \sim q(\mathbf{x})$.

For $i > 2$, members of $\mathcal{B}_M^{i-1}$ are drawn:

$$\mathbf{z}_j^{i-1} \sim q_\phi(\mathbf{z}^{i-1}|\mathbf{z}_j^{i-2}, \mathbf{x}_j) \tag{36}$$

and for $i = 2$:

$$\mathbf{z}_j^1 \sim q_\phi(\mathbf{z}^1|\mathbf{x}_j) \tag{37}$$

Thus each member of this batch $\mathcal{B}_M^{i-1}$ is the descendant of a particular datapoint that was sampled in an iid minibatch $\mathcal{B}_M$ as defined above. We similarly define $r(\mathcal{B}_M^{i-1}|\mathbf{z}^{i-1}, \mathbf{x})$ as the probability of selecting a particular minibatch $\mathcal{B}_M^{i-1}$ of these values out from our set $\{(\mathbf{x}_n, \mathbf{z}_n^{i-1})\}$ (of cardinality $N$) given that we have selected into our minibatch one particular pair of values $(\mathbf{x}, \mathbf{z}^{i-1})$ from these $N$ values. Like above, $r(\mathcal{B}_M^{i-1}|\mathbf{z}^{i-1}, \mathbf{x}) = \frac{1}{N}^{M-1}$

Now we can consider $\mathbb{E}_{q_\phi(\mathbf{z}^i)} \log q_\phi(\mathbf{z}^i)$ for $i > 1$:

$$\mathbb{E}_{q_\phi(\mathbf{z}^i)} \log q_\phi(\mathbf{z}^i) = \mathbb{E}_{q_\phi(\mathbf{z}^i,\mathbf{z}^{i-1},\mathbf{x})} [\log \mathbb{E}_{q_\phi(\mathbf{z}^{i-1},\mathbf{x})} [q_\phi(\mathbf{z}^i|\mathbf{z}^{i-1},\mathbf{x})]] \tag{38}$$

$$= \mathbb{E}_{q_\phi(\mathbf{z}^i,\mathbf{z}^{i-1},\mathbf{x})} [\log \mathbb{E}_{p(\mathcal{B}_M^{i-1})} [\frac{1}{M} \sum_{m=1}^{M} q_\phi(\mathbf{z}^i|\mathbf{z}_m^{i-1},\mathbf{x}_m)]] \tag{39}$$

$$\geq \mathbb{E}_{q_\phi(\mathbf{z}^i,\mathbf{z}^{i-1},\mathbf{x})} [\log \mathbb{E}_{r(\mathcal{B}_M^{i-1}|\mathbf{z}^{i-1},\mathbf{x})} [\frac{p(\mathcal{B}_M^{i-1})}{r(\mathcal{B}_M^{i-1}|\mathbf{z}^{i-1},\mathbf{x})} \frac{1}{M} \sum_{m=1}^{M} q_\phi(\mathbf{z}^i|\mathbf{z}_m^{i-1},\mathbf{x}_m)]] \tag{40}$$

$$= \mathbb{E}_{q_\phi(\mathbf{z}^i,\mathbf{z}^{i-1},\mathbf{x})} [\log \mathbb{E}_{r(\mathcal{B}_M^{i-1}|\mathbf{z}^{i-1},\mathbf{x})} [\frac{1}{NM} \sum_{m=1}^{M} q_\phi(\mathbf{z}^i|\mathbf{z}_m^{i-1},\mathbf{x}_m)]] \tag{41}$$

Where we have followed the same steps as in the previous subsection.

During training, one samples a minibatch $\{\mathbf{z}_1^{i-1}, \mathbf{z}_2^{i-1}, ..., \mathbf{z}_M^{i-1}\}$, where each is constructed by sampling ancestrally. Then one can estimate $\mathbb{E}_{q_\phi(\mathbf{z}^i)} \log q_\phi(\mathbf{z}^i)$ as:

$$\mathbb{E}_{q_\phi(\mathbf{z}^i)} \log q_\phi(\mathbf{z}^i) \approx \frac{1}{M} \sum_{k=1}^{M} [\log \sum_{j=1}^{M} q_\phi(\mathbf{z}_k^i|\mathbf{z}_j^{i-1},\mathbf{x}_j) - \log NM] \tag{42}$$

and $\mathbf{z}_k^i$ is a sample from $q_\phi(\mathbf{z}^i|\mathbf{z}_k^{i-1},\mathbf{x}_k)$. In our approach we only need terms of this form for $i = L$, so we have:

$$\mathbb{E}_{q_\phi(\mathbf{z}^L)} \log q_\phi(\mathbf{z}^L) \approx \frac{1}{M} \sum_{k=1}^{M} [\log \sum_{j=1}^{M} q_\phi(\mathbf{z}_k^L|\mathbf{z}_j^{L-1},\mathbf{x}_j) - \log NM] \tag{43}$$

and $\mathbf{z}_k^L$ is a sample from $q_\phi(\mathbf{z}^L|\mathbf{z}_k^{L-1},\mathbf{x}_k)$.

# D SEATBELT-VAE RESULTS

## D.1 SEATBELT-VAE LAYERWISE ATTACKS

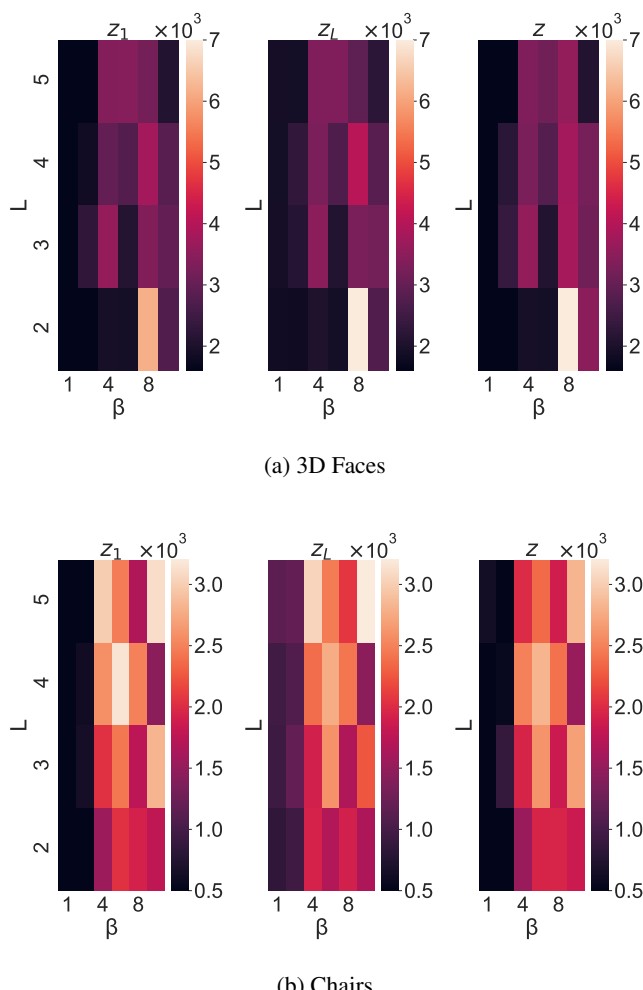

(a) 3D Faces

(b) Chairs

Figure D.8: $-\log p_\theta(\mathbf{x}^t|\tilde{\mathbf{z}})$, $\tilde{z} \sim q(\mathbf{z}|\mathbf{x}+d)$ where $d$ is some adversarial distortion, for Seatbelt-VAEs trained on (a) 3D Faces and (b) Chairs; over $\beta$ and $L$ values for *latent* attacks. We attack the bottom layer $(\mathbf{z}^1)$, the top layer $(\mathbf{z}^L)$, and finally show the effect when attacking all layers $(\mathbf{z})$. Larger values of $-\log p_\theta(\mathbf{x}^t|\tilde{\mathbf{z}})$ correspond to less successful adversarial attacks. Generally attacking all layers seems to give the attacker a slight advantage (as seen by the slightly lower $-\log p_\theta(\mathbf{x}^t|\tilde{\mathbf{z}})$ values for Faces and Chairs).

## D.2 SEATBELT-VAE ATTACKS BY MODEL DEPTH AND $\beta$

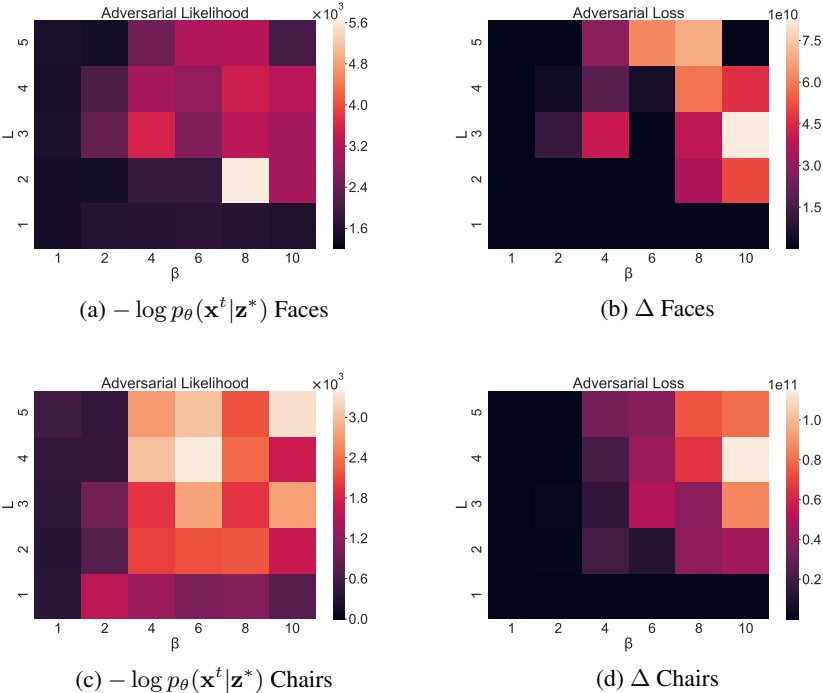

Figure D.9: Here we measure the robustness of TC-penalised models numerically. Sub-figures (a) and (c) show $-\log p_\theta(\mathbf{x}^t|\mathbf{z}^*)$, the adversarial likelihood of a target image $\mathbf{x}^t$ given an attacked latent representation $\mathbf{z}^*$ for Seatbelt-VAEs for Chairs and 3D Faces. Larger likelihood values correspond to less successful adversarial attacks. Sub-figures (b) and (d) show adversarial loss $\Delta$ for Seatbelt-VAEs for Chairs and 3D Faces. We show these likelihood and loss values over $\beta$ and $L$ (total number of stochastic layers) values for attacks. Note that the bottom rows of all figures have $L = 1$, and thus correspond to $\beta$-TCVAEs. The leftmost column corresponds to models with $\beta = 1$, which are vanilla VAEs and hierarchical VAEs. As we go to the largest values of $\beta$ and $L$ for both Chairs and 3D Faces, $\Delta$ grows by a factor of $\approx 10^7$ and $-\log p_\theta(\mathbf{x}^t|\mathbf{z}^*)$ doubles. These results tell us that depth and TC-penalisation together, i.e Seatbelt-VAE, can offer immense protection from the adversarial attacks studied.

# E AGGREGATE ANALYSIS OF ADVERSARIAL ATTACK

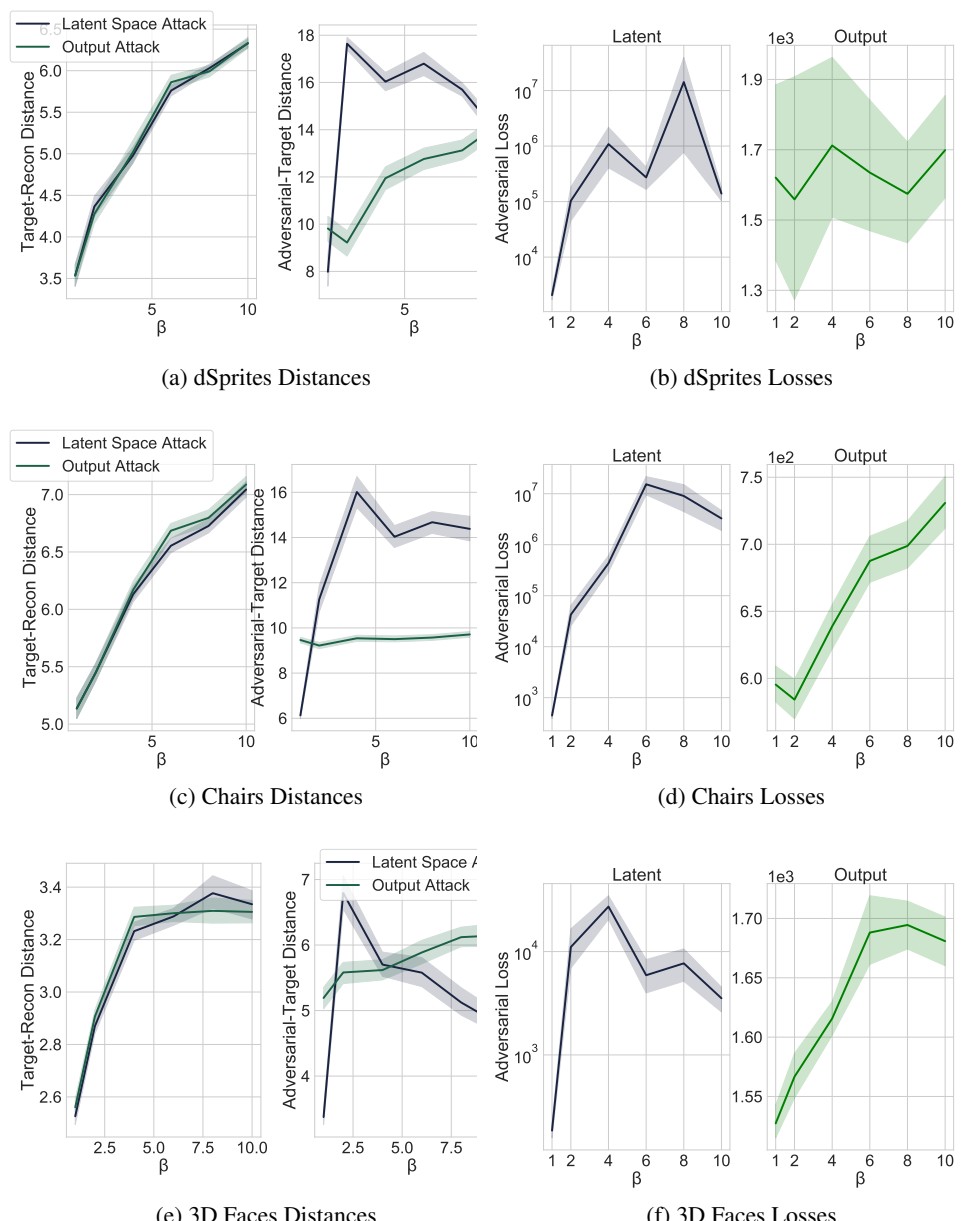

Figure E.10: Plots showing the effect of varying $\beta$ in a $\beta$-TCVAE trained on dSprites (a,b), Chairs (c,d), and 3D Faces (d,e) on: the $L_2$ distance from the adversarial target $x^t$ to its reconstruction when given as input (target-recon distance) and the $L_2$ distance between the adversarial input $x^*$ and $x^t$ (adversarial-target distance); and the adversarial objectives $\Delta$. We also include these metrics for "output" attacks Gondim-Ribeiro et al. (2018), which we find to be generally less effective. In such attacks the attacker directly tries to reduce the L2 distance between the reconstructed output and the target image. For latent attacks the adversarial-target $L_2$ distance grows more rapidly than the target-recon distance (i.e the degradation of reconstruction quality) as we increase $\beta$. This effect is much less clear for output attacks. This makes it apparent that the robustness we see in $\beta$-TCVAE to latent space adversarial attacks is not due the degradation in reconstruction quality we see as $\beta$ increases. It is also apparent that increasing $\beta$ increases the adversarial loss for latent attacks and output attacks.

## E.1 DISENTANGLING AND ROBUSTNESS?

Although we are using regularisation methods that were initially proposed to encourage disentangled representations, we are interested here in their effect on robustness *not* whether the representations we learn are in fact disentangled. This is not least due to the questions that have arisen about the hyperparameter tuning required for disentangled representations Locatello et al. (2019); Rolinek et al. (2019). For us the $\beta$ pre-factor is just the degree of regularisation imposed.

However, it may be of interest to see what relationship, if any, exists between the ease of attacking of a model and how disentangled it is. Here we show the MIG score (Chen et al., 2018) against the achieved adversarial loss on the Faces data for $\beta$-TCVAEs. MIG measures the degree to which representations are disentangled and larger adversarial losses correspond to a less successful attack. Shading is over the range of $\beta$ and $d_z$ values. There does not seem to be any simple correspondence between increased MIG and increases in adversarial loss, indicative of a less successful attack.

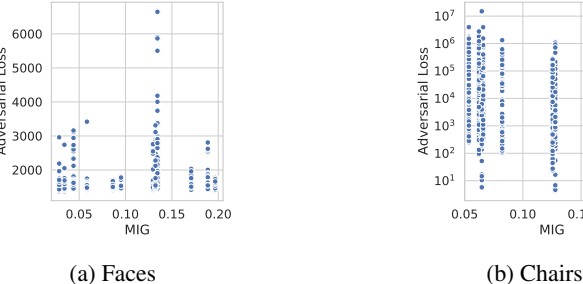

(a) Faces       (b) Chairs

Figure E.11: Adversarial attack loss reached vs MIG score for $\beta$-TCVAEs trained on Faces and Chairs presented for a range of $\beta = \{1, 2, 4, 6, 8, 10\}$ and $d_z = \{8, 32\}$ values.

# F    ROBUSTNESS TO NOISE

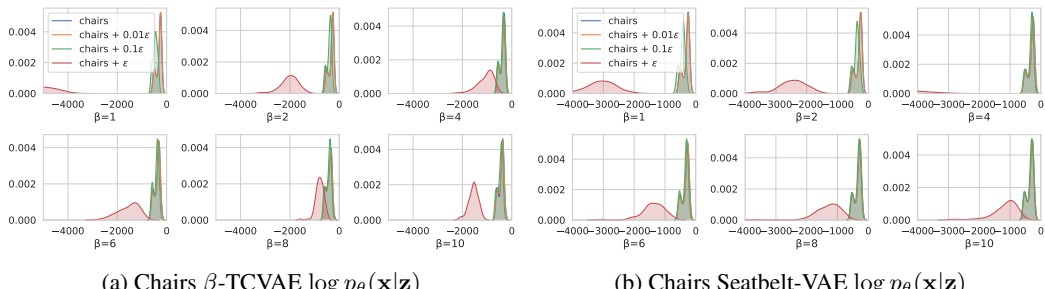

(a) Chairs $\beta$-TCVAE $\log p_\theta(\mathbf{x}|\mathbf{z})$  (b) Chairs Seatbelt-VAE $\log p_\theta(\mathbf{x}|\mathbf{z})$

Figure F.12: Here we measure the robustness of both $\beta$-TCVAE and Seatbelt-VAE when Gaussian noise is added to Chairs. Within each plot a range of $\beta$ values are shown. We evaluate each model's ability to decode a noisy embedding to the original non-noised data $\mathbf{x}$ by measuring the distribution of $\log p_\theta(\mathbf{x}|\mathbf{z})$ when $\mathbf{z} \sim q_\phi(\mathbf{z}|\mathbf{x} + a\boldsymbol{\epsilon})$ ($a$ some scaling factor taking values in $\{0.1, 0.5, 1\}$ and $\boldsymbol{\epsilon} \sim \mathcal{N}(0,1)$) for which higher values indicate better denoising. We show these likelihood values as density plots for the $\beta$-TCVAE in (a) and for the Seatbelt-VAE with $L = 4$ in (b), taking $\beta \in \{1, 2, 4, 6, 8, 10\}$. Note the axis scalings are different for each subplot. We see that for both models using $\beta > 1$ produces autoencoders that are better at denoising their inputs. Namely, the mean of the density, i.e. $\mathbb{E}_{q_\phi(\mathbf{z}|\mathbf{x}+\boldsymbol{\epsilon})}[\log p_\theta(\mathbf{x}|\mathbf{z})]$, shifts dramatically to higher values for $\beta > 1$ relative to $\beta = 1$. In other words, for both these models, the likelihood of the dataset in the noisy setting is much closer to the non-noisy dataset when $\beta > 1$ across all noise scales ($0.1\boldsymbol{\epsilon}$, $0.5\boldsymbol{\epsilon}$, $\boldsymbol{\epsilon}$).

## G  IMPLEMENTATION DETAILS

All runs were done on the Azure cloud system on NC6 GPU machines.

### G.1  ENCODER AND DECODER ARCHITECTURES

We used the same convolutional network architectures as Chen et al. (2018). For the encoders of all our models ($q(\cdot|\mathbf{x})$) we used purely convolutional networks with 5 convolutional layers. When training on single-channel (binary/greyscale) datasets such as dSprites, 3D Faces, or Chairs the 5 layers took the following number of filters in order: $\{32, 32, 64, 64, 512\}$. For more complex RGB datasets, such as CelebA, the layers had the following number of filters in order: $\{64, 64, 128, 128, 512\}$. The mean and variance of the amortised posteriors are the output of dense layers acting on the output of the purely convolutional network, where the number of neurons in these layers is equal to the dimensionality of the latent space $\mathcal{Z}$.

Similarly, for the decoders ($p(\mathbf{x}|\mathbf{z})$) of all our models we also used purely convolutional networks with 6 deconvolutional layers. When training on single-channel (binary/greyscale) datasets, dSprites, 3D Faces, or Chairs, the 6 layers took the following number of filters in order: $\{512, 64, 64, 32, 32, 1\}$. For CelebA the layers had the following number of filters in order: $\{512, 128, 128, 64, 64, 3\}$. The mean of the likelihood $p(\mathbf{x}|\cdot)$ was directly encoded by the final de-convolutional layer. The variance of the decoder, $\sigma$, was fixed to 0.1.

For $\beta$-TCVAE the range of $d_{\mathbf{z}}$ values used was $\{4, 6, 8, 16, 32, 64, 128\}$. For Seatbelt-VAEs the number of units in each layer $\mathbf{z}^i$ decreases sequentially. There is a list z_sizes for each dataset, and for a model of $L$ layers that the last $L$ entries to give $d_{\mathbf{z},i}, i \in \{1, ..., L\}$.

$$\{d_{\mathbf{z}}\}^{\mathrm{dSprites}} = \{96, 48, 24, 12, 6\} \tag{44}$$

$$\{d_{\mathbf{z}}\}^{\mathrm{Chairs}} = \{96, 48, 24, 12, 6\} \tag{45}$$

$$\{d_{\mathbf{z}}\}^{\mathrm{3DFaces}} = \{96, 48, 24, 12, 6\} \tag{46}$$

$$\{d_{\mathbf{z}}\}^{\mathrm{CelebA}} = \{256, 128, 64, 32\} \tag{47}$$

For Seatbelt-VAEs we also have the mappings $q_\phi(\mathbf{z}^{i+1}|\mathbf{z}^i, \mathbf{x})$ and $p_\theta(\mathbf{z}^i|\mathbf{z}^{i+1})$. These are amortised as MLPs with 2 hidden layers with batchnorm and Leaky-ReLU activation. The dimensionality of the hidden layers also decreases as a function of layer index $i$:

$$d_{\mathbf{h}}(q_\phi(\mathbf{z}^{i+1}|\mathbf{z}^i, \mathbf{x})) = \mathrm{h}_{\mathrm{sizes}}[i] \tag{48}$$

$$d_{\mathbf{h}}(p_\theta(\mathbf{z}^i|\mathbf{z}^{i+1})) = \mathrm{h}_{\mathrm{sizes}}[i] \tag{49}$$

$$\mathrm{h}_{\mathrm{sizes}} = [1024, 512, 256, 128, 64] \tag{50}$$

To train the model we used ADAM Kingma & Lei Ba (2015) with default parameters, a cosine decaying learning rate of 0.001, and a batch size of 1024. All data was pre-processed to fall on the interval -1 to 1. CelebA and Chairs were both downsampled and cropped as in Chen et al. (2018) and Kulkarni et al. (2015) respectively. We find that using *free-bits* regularisation (Kingma et al., 2016) greatly ameliorates the optimisation challenges associated with DLGMs.

