# OpenReview forum: "Improving VAEs' Robustness to Adversarial Attack"
_ICLR.cc/2021/Conference — ICLR 2021 Poster_

### Official Review · AnonReviewer2 · 2020-10-23

**Rating:** 7
**Confidence:** 2

**Review:**

Summary: This work builds on the vulnerability of VAEs to adversarial attacks to propose investigate how training with alternative losses may alleviate this problem, with a specific focus on disentanglement. In particular it is found that disentanglement constraints may improve the robustness to adversarial attacks, to the detriment of the performance. In order to get the best of both, the author(s) propose a more flexible (hierarchical) model, trained with the beta-TC penalization on the ELBO. The algorithm, named Seatbelt-VAE, shows improvement over the beta-TC VAE in terms of reconstruction, as well as in term of adversarial robustness for several datasets (Chairs, 3D Faces, dSprites).

Comments:
1. The paper is well-written and the underlying reasoning is easy to follow.
2. The experiments are sound and well documented (results are reported across latent space dimensions, and adversarial attack parameters)

Questions:
1. I am wondering how the bias of estimating the TC term on Z^L in Eq (8) scales with L and the minibatch size, compared to the more simple TC estimator from Chen et al. (2018) and if the author(s) had any evidence from the experiments that it might be problematic? Does the algorithm require even larger batch sizes?
2. Should this approach be compared as well to weight decay or other simple regularization on the weights?

Minor questions:
3. I wish the paper would make a stronger connection between disentanglement and robustness. The beta-TC VAE is only one choice among other possible to constrain the variational network. Did the authors ever try anything else?
4. Is it possible that the TC-VAE is effective at providing a defense against adversarial attacks in this manuscript because of the nature of the attack used in this manuscript (Eq. (1))? If the attack was not based on the Kullback-Leibler divergence, but based on another geometry, maybe another disentanglement constraints would be more performant?

---

> ### Author Response · Authors · 2020-11-14
> **Response to Reviewer #2**
>
> Thank you for your thoughtful review.  We are glad you liked the paper and in particular its clarity and experiments. We have uploaded a significant update to the paper to address the concerns raised, including a number of new experiments.
>
> ----------------------
>
> “1. I am wondering how the bias of estimating the TC term on Z^L in Eq (8) scales with L and the minibatch size, compared to the more simple TC estimator from Chen et al. (2018) and if the author(s) had any evidence from the experiments that it might be problematic? Does the algorithm require even larger batch sizes?“
>
>
> We expect the bias in this estimator to be similar to that in the single-layer case, i.e. to be roughly constant with L. This is because in the derivation in Appendix C the samples from the layers below act like the sampled data in the single-layer case (see the text starting "During training," in C.2). Large minibatch sizes are required (as small minibatches produce large biases, see Appendix C Mathieu et al 2019), but no larger than those of Chen et al. (2018): we use a size of 1024 which is the same as they use in the majority of their experiments.
>
> ----------------------
>
> “2. Should this approach be compared as well to weight decay or other simple regularization on the weights?“
>
>
> Though we believe such comparisons could be quite interesting, we believe they are beyond the scope of this paper as nobody has considered using such an approach as a means to achieve robust VAEs before, so any related baseline would be a new contribution in on itself.  Moreover, there is nothing stopping one using such regularisations alongside our methods as well: our approach is agnostic to the neural network architecture of a given VAE. Further, any weight decay or similar method of regularisation would both forgo a neat probabilistic interpretation and it is not immediately obvious to us how they would be calibrated to give the required robustness in the resulting model. This would make a comparison with overlap-controlling methods, like TC-penalisation, very difficult.
>
> ----------------------
>
> “3. I wish the paper would make a stronger connection between disentanglement and robustness.“
>
>
> We have expanded and improved our discussion of the links between methods proposed for obtaining disentangled representations and robustness to adversarial attack in Sec 3, and in Appendix E.1. We note that we are repurposing methods originally used for disentangling, because of their regularising properties on the latent space, to induce robust VAEs. We are not claiming that disentangled representation induce more robust VAEs, and indeed in the new Figure E.10 we can see that disentangling scores are not correlated with adversarial robustness.
>
> ----------------------
>
> “The beta-TC VAE is only one choice among other possible to constrain the variational network. Did the authors ever try anything else?“
>
>
> Some experiments where initially run using the β-VAE but were quickly dropped because of the poor reconstruction performance.  More precisely, in Section 3 we compare the β-TCVAE to the β-VAE and found that the β-TCVAE had more favourable properties in terms of reconstruction quality and noisier latent embeddings. The degradation in model quality that β-VAE type penalisations induce render them much less useful for downstream tasks, effectively ruling them out from our consideration.  Other ways of constraining the variational network have less direct links to the overlap we are advocating controlling, which is why we did not consider them.  However, we agree looking into these could make for interesting future work.
>
> ----------------------
>
> “4. Is it possible that the TC-VAE is effective at providing a defense against adversarial attacks in this manuscript because of the nature of the attack used in this manuscript (Eq. (1))? If the attack was not based on the Kullback-Leibler divergence, but based on another geometry, maybe another disentanglement constraints would be more performant?”
>
>
> This is an interesting question. To explore this we have added two new attacks to the paper (see Figure 5). The first, proposed in Kos et al. 2018, is to attack the output of the VAE: to try to maximise the ELBO of the target under the distorted input. The second, a new attack stimulated by this comment, is to attack the 2-Wasserstein distance between the target posterior and the attacked posterior. These experiments highlight that TC-penalisation can protect against attacks based on geometries other than the KL. It is possible that different latent regularisation methods will be most effective against a specific geometry and we think that exploring this regularisation-geometry connection would be an interesting follow-up to this work.

---

> > ### Comment · AnonReviewer2 · 2020-11-23
> > **Thank you**
> >
> > I would like to thank the authors for answering my questions. I have kept my score.

---

### Official Review · AnonReviewer1 · 2020-10-27
**Hierarchical disentangled VAE for adversarial robustness.**

**Rating:** 6
**Confidence:** 5

**Review:**

The paper considers the regularization of latent space toward achieving adversarial robustness against latent space attack. The paper demonstrates the applicability of disentanglement promoting VAEs for achieving adversarial robustness and further enhancing such VAEs by considering their hierarchical counterparts. The paper demonstrates their results in the benchmark datasets considered in the disentanglement and computer vision literature. The overall research direction pursued by this paper is exciting. However, I have some concerns, which include:

1. The paper attempts to establish the connection between disentanglement and robustness. The linkage, however, is not clear. In section 3, the paper argues for the smoothness of the encoder mapping and the decoder mapping. Toward this, the paper postulates additional regularization to enforce "simplicity" or "noiseness". First of all, it is unclear how disentangled latent space helps achieve "simplicity" in the "encode-decode process". Secondly, regarding "noiseness", it is not explained what extra would disentangled version of VAEs (e.g., TCVAE) provide compared to the standard setup.

2. In section 3.2, the paper empirically demonstrates the connection between disentanglement and adversarial robustness. However, the evaluation carried out are not explicit. Firstly, to demonstrate the connection, the paper uses the attacker's achieved loss \delta (from Eqn 1) as the metric. Although the \delta is shown across different \beta values, it is still unclear if disentanglement is directly related to robustness. Can the authors point out some disentanglement metrics (e.g., MIG) for each beta and compare MIG vs. \delta? Also, the curves are combined for all the d_z. What is the motivation behind doing that? Because it has been known that disentanglement behavior is related to the dimension of latent space. Also, authors could consider decomposing the first term of \delta for all the latent space dimensions and analyze if the disentangled dimensions are robust compared to the entangled dimension. This could be more helpful to establish the linkage. Secondly, authors have picked TCVAE considering "reconstruction quality" compared to \beta-VAE, but in Fig 2 (right), ELBO is compared. Can the authors compare the reconstruction error? Also, for fig 3, I think it is natural to see the comparison with \beta-VAE. Why is such a comparison not included?

3. In section 4, for the motivation for the hierarchical TC-penalised VAEs, the paper states that "TC-penalisation in single layer VAEs comes at the expense of model reconstruction quality". However, this directly contradicts the use of TC-VAE in the previous section. Although the results presented afterward support the authors' statement, the motivation must be clear and well written. The same comments for section 3.2 apply here too.

4. The experimental results demonstrating protection against downstream tasks is performed using a simple 2-lear MLPs. This is different from the regular CNN network commonly considered for these datasets. Although this was meant to demonstrate the proposed model's efficacy, it would be more clear if the experimental setup is consistent with the current literature setting. Also, can the authors point out the initial results for the models before the attack?


Minor comments:

- There are a lot of grammatical errors and hard-to-follow sentences. Some examples:
    - ".. are not only even more .."
    - ".. attack the models using methods outlined ..". But Eq (1) refers to only one method, right?
    - "… then \delta too is small .."

(Update): The score has been updated after a rebuttal from the authors.

---

> ### Author Response · Authors · 2020-11-14
> **Response to Reviewer #1, part 1**
>
> Thank you for your thoughtful review.  We are glad you felt that the paper takes an exciting research direction. We have uploaded a significant update to the paper to address the concerns raised, including a number of new experiments.
>
> ----------------------
>
> “1. The paper attempts to establish the connection between disentanglement and robustness. The linkage, however, is not clear... it is unclear how disentangled latent space helps achieve "simplicity" in the "encode-decode process"... regarding "noiseness", it is not explained what extra would disentangled version of VAEs (e.g., TCVAE) provide compared to the standard setup."
>
>
> We think there may have been a bit of a confusion here: as explained in our general response to all reviewers, we are not trying to establish a direct link between disentangled representations and robustness, but instead that regularisers that were proposed for disentangling can be reliably used to confer robustness to VAEs.  We have re-written Section 3 and 3.1 to make these connections clearer. In particular, we wish to highlight that we are not saying that it is disentangled representations themselves that are most robust, but rather that methods first proposed in the disentangling literature can induce more robust models by way of their regularising properties.
>
> As we demonstrate in Figure 2, methods such as the Beta-VAE and the Beta-TCVAE increase the variance of encoder embeddings effectively increasing the ‘noisiness’ of the encoding process. When one increases the variance of embeddings, there is greater ‘overlap’ between the different per-datapoint posteriors as highlighted by Mathieu et al. 2019. When there is no overlap (ie no noise), VAEs can default to a lookup-table behaviour where there is no direct pressure for similar encodings to correspond to similar images. Lookup table behaviour is very vulnerable to attack in that small changes in the input from an adversary can lead to large disruptions in neural network outputs. Adding noise forces the VAE to smooth the encode-decode process in that similar images will lead to similar embeddings in the latent space (see Mathieu et al. 2019 for further discussion of this), ensuring that small changes in the input result in small changes in the latent space and those changes result in small changes in the decoded outputs. This proportional input-output change is what we refer to as a ‘simple’ encode-decode process.  One can already see how it provides adversarial robustness: small changes in the input result in small changes in the reconstructed output and an adversary needs to effect a much larger disruption to the input to enact a change in the output.
>
> ----------------------
>
> “2. ...the paper empirically demonstrates the connection between disentanglement and adversarial robustness. However, the evaluation carried out are not explicit... it is still unclear if disentanglement is directly related to robustness. Can the authors point out some disentanglement metrics (e.g., MIG) for each beta and compare MIG vs. \delta?”
>
>
> As explained above, we are not arguing that it is disentangled representations per se that confer robustness. However, we felt the suggested experiment was an excellent thing to try in order investigate whether there are indeed connections.  To this end, we added a new experiment in Figure E.10 where we plot Δ against MIG for beta-TCVAEs (note that disentangling metrics have not been defined for hierarchical VAEs such that a similar comparison is not possible for Seatbelt-VAEs).  Interestingly, we see no clear relationship between these two quantities, suggesting that there is indeed not a directly link between disentanglement and robustness. The reason for this is that while the regularisation was reliably able to induce robustness, achieving disentanglement is substantially more finicky: as shown by Locatello et al. 2019, one requires very careful hyperparameter selection and a fair degree of luck to learn disentangled representations.  Because we did not perform the parameter tuning and repeat runs to get around this, the setups we are using do not reliably produce disentangled representations, even though they do reliably produce robust ones.

---

> > ### Author Response · Authors · 2020-11-14
> > **Response to Reviewer #1, part 2**
> >
> > ----------------------
> >
> > “Also, the curves are combined for all the d_z. What is the motivation behind doing that? [It is] known that disentanglement behavior is related to the dimension of latent space.”
> >
> >
> > We hope our response above has clarified that we are not investigating the relationship between disentangled representations and adversarial robustness. Yes, given the results in Locatello et al. (2019) and Rolinek et al. (2019) we can expect that changes in d_z would lead to changes in how disentangled our learnt representations are. However, as we are not relying on obtaining disentangled representations for our method to work, rather on the regularisation that TC-penalisation provides, for us the range of d_z values we consider demonstrates that our proposal works within a broad range of network hyperparameters.  This also further explains why we do not see a clear relationship between the MIG and Δ for the experiment in Figure E.10.
> >
> > ----------------------
> >
> > “Authors have picked TCVAE considering "reconstruction quality" compared to \beta-VAE, but in Fig 2 (right), ELBO is compared. Can the authors compare the reconstruction error?"
> >
> >
> > We have added a subfigure in Figure 2 confirming the improved fidelity of β-TCVAE reconstructions and that these follow the same trends as the ELBO: as β is increased the log likelihood of the input data decreases much more briskly for β-VAEs than for β-TCVAEs, so the β-VAE curve is below that for β-TCVAEs.
> >
> > ----------------------
> >
> > “For fig 3, I think it is natural to see the comparison with \beta-VAE. Why is such a comparison not included?“
> >
> >
> > In short, because the reconstructions and ELBO of the β-VAE are already so poor (cf Figure 2) that they cannot produce VAEs that are useful in downstream tasks except at very small values of β, even if they are robust to attack.  This, coupled with the fact that TC penalisation offers greater overlap as per Figure 2 [left], meant we decided to focus on TC-penalised models.  We are happy to add this comparison in if you think it is important, but we decided to prioritise the other new experiments and these responses first.
> >
> > ----------------------
> >
> > “3. ... the paper states that "TC-penalisation in single layer VAEs comes at the expense of model reconstruction quality". However, this directly contradicts the use of TC-VAE in the previous section. Although the results presented afterward support the authors' statement, the motivation must be clear and well written. The same comments for section 3.2 apply here too."
> >
> >
> > We agree that this wording could have been clearer and have now updated it: as you say there is still a quality-regularisation tradeoff for TC-VAEs, we are simply saying that it tends to achieve better tradeoffs than β-VAEs. The Seatbelt-VAE improves this tradeoff even further by bringing TC-penalisation to the hierarchical setting.
> >
> > ----------------------
> >
> > “4. The experimental results demonstrating protection against downstream tasks is performed using a simple 2-lear MLPs. This is different from the regular CNN network commonly considered for these datasets. Although this was meant to demonstrate the proposed model's efficacy, it would be more clear if the experimental setup is consistent with the current literature setting. Also, can the authors point out the initial results for the models before the attack?”
> >
> >
> > We agree with this suggestion and have included results in Table 1 that look at the performance of convolutional classifiers trained on the reconstructions of VAEs. We find that as before Seatbelt-VAEs protects the accuracy of these models the most. Note that the initial accuracies are simply the value in the table minus the value in parentheses, which is the observed degradation in accuracy (e.g. for the top left value the initial accuracy is 0.17-(-0.35)=0.52).
> >
> > ----------------------
> >
> > “Minor Comments”
> >
> > We thank the reviewer for pointing out these mistakes and have rectified these sentences accordingly to improve clarity.

---

> > > ### Comment · AnonReviewer1 · 2020-11-21
> > > **Thanks for the updates.**
> > >
> > > Thanks for the detailed response and the care with which you addressed my concerns. I really appreciate it! I agree that there was some confusion on my end regarding this work and the disentangling. This confusion is sorted now.

---

> > > > ### Author Response · Authors · 2020-11-22
> > > > **Any remaining questions?**
> > > >
> > > > Thanks for following up! It is great to hear that the confusions have been cleared up and that you appreciated our experimental and writing updates.
> > > >
> > > > We noticed that you mentioned the following in your original comment:
> > > >
> > > > _"Considering the experimental and writing updates made to the paper, I have increased my initial score."_
> > > >
> > > > But that you have since removed this and have not updated your score.  We were thus wondering if you still planned to do this?  If you have changed your mind, is there anything more you would like to see or that you feel requires further clarification?
> > > >
> > > > As you mentioned yourself, we have put a lot of effort into addressing your concerns and have noticeably improved the paper (including new experiments for two of the four concerns raised), so we really hope you do consider a score increase!
> > > >
> > > > Thanks again for your time and consideration.

---

> > > > > ### Comment · AnonReviewer1 · 2020-11-22
> > > > > **Update.**
> > > > >
> > > > > Yes, I plan to increase the initial score after the end of this author-reviewer discussion phase. All of my comments have been addressed, and I am currently studying other reviewers' comments and the authors' rebuttal. I hope this answers your queries.

---

> > > > > > ### Author Response · Authors · 2020-11-22
> > > > > > **Thank you**
> > > > > >
> > > > > > Wonderful, thanks for clarifying! Do let us know if there is anything else we can help with.

---

### Official Review · AnonReviewer4 · 2020-10-28
**Paper is well organized, but there are issues in experimental results.**

**Rating:** 6
**Confidence:** 3

**Review:**

--Summary:
They proposed a robust method for the adversarial attack on VAE using a hierarchical version of $\beta$-TCVAE and conduct analysis on the relationship between disentanglement and robustness to support their choice of approach. The experimental results demonstrate the effectiveness of the proposed defense method.

--Strongness:
1. The paper is well organized.
2. They provide extensive analysis of their approach and provide proof.
3. They demonstrate that the proposed method is more robust to other VAE baselines for the attacks.

--Weakness:
1. The qualitative results are not quite convincing. See below question 1.


--Questions:
1. Question on qualitative results:
- For Figure 1, the adversarial examples for the three approaches are largely different. For the baselines -- VAE and $\beta$-TCVAE, the provided inputs look not like just applying a small noise/distortion (which is the setting of adversarial attack) but have a huge difference from the original input. Therefore, the reconstructions can be expected that it won't look similar to the original reconstruction.
- In Figure 6(c), are you using similar adversarial examples / similar amount of distortions to generate the reconstruction images for $\beta$-TCVAE and your approach?

I'm not an expert in the adversarial attack domain, but shouldn't the adversarial examples be similar across different baselines?

2. Figure 4(b) top-right adversarial example seems not to be distorted (almost the same as the input). Is it the adversarial example derived by applying distortion? Since the distortion is large, I don't know why the adversarial looks just the same as the input.

--Recommendation
In sum, the paper is well organized and consists of extensive theoretical proofs and experiments. While I'm not very convinced by the results. I vote for a more neutral score for now. The authors are couraged to address these issues.

---

> ### Author Response · Authors · 2020-11-14
> **Response to Reviewer #4**
>
> Thank you for your thoughtful review.  We are glad you like the paper and thank you for highlighting the strengths of our work. We have uploaded a significant update to the paper to address the concerns raised, including a number of new experiments.
>
> ----------------------
>
> “For Figure 1, the adversarial examples for the three approaches are largely different...Therefore, the reconstructions can be expected that it won't look similar to the original reconstruction.“
>
>
> You are correct that the learnt perturbation is for the Seatbelt-VAE is smaller than for the vanilla VAE or β-TCVAE, but this actually because of the attack struggling, rather than because of any constraints on the allowed perturbation. Recall that here the adversary is trying to reconstruct to some target image, not to simply undermine the reconstruction of the original; increasing the size of the perturbation is not always helpful for this goal. Here we are following exactly the method of Gondim-Ribeiro et al. (2018) in how we plot these attacks and the inputs shown correspond to the most adversarial input the attacker was able to find. The reason that the Seatbelt-VAE input only undergoes a small perturbation is because it is sufficiently robust that (even when λ is almost zero) the attacker is not able to make the reconstruction look more like the target image in any meaningful way, such that it optimiser never drifts far from the initial input.  This is in contrast to the vanilla-VAE where the attack is able to move through the latent space and find a perturbation that reconstructs to the adversary’s target image. We note that the β-TCVAE is also mostly robust here, as here the attacker is unable to induce the desired adversarial reconstruction, even though the learnt attack may be of large magnitude (but it is still able to find a perturbation that produces a reconstruction more like the target than the original, unlike Seatbelt where the adversary is effectively completely stuck). We have clarified this in the text.
>
> ----------------------
>
> “In Figure 6(c), are you using similar adversarial examples / similar amount of distortions to generate the reconstruction images for β-TCVAE and your approach?”
>
>
> In Figure 6 we are simply showing reconstructions from our models — no attacks are being done. Our aim in Figure 6(c) is to show that qualitatively, the Seatbelt-VAEs are better able to reconstruct images than β-TCVAEs for an equivalent β penalisation. This highlights that Seatbelt models can provide a better trade-off between robustness and model quality.
>
> ----------------------
>
> "Shouldn't the adversarial examples be similar across different baselines?...  Figure 4(b) top-right adversarial example seems not to be distorted... Is it the adversarial example derived by applying distortion? Since the distortion is large, I don't know why the adversarial looks just the same as the input."
>
>
> In Figure 4 all attack plots including 4.b) are rescaled to [0-1] for ease of viewing, so although the attack might appear to have large magnitude it is in reality of quite small scale (the adversarial example is indeed derived by adding the distortion). We realise this is potentially confusing and have made sure to address this in the text. We also note that, as mentioned in response to the reviewer’s previous query, the small perturbation plotted here is one of the best attacks the adversary could generate and is a marker of the robustness of Seatbelt-VAEs. The attacker was unable to learn an attack that reconstructs to the target image for a large range of attack norm penalisations, whereas for the vanilla-VAE the attacker is able to do so.

---

### Official Review · AnonReviewer3 · 2020-10-31
**Suggests connections between robustness and disentangled representations, but isn't explored fully**

**Rating:** 7
**Confidence:** 3

**Review:**

The paper considers the problem of training VAEs which are robust to adversarial attacks. It shows that learning disentangled representations improves the robustness of VAE. However, this hurts the reconstruction accuracy. The paper then shows that using hierarchical VAEs can ensure robustness without sacrificing reconstruction.

Strengths:

1. The problem considered is interesting and relevant. There is very little work on training robust VAEs---though they have been found to not be robust, and are also used for training robust classifiers downstream.
2. The paper uncovers some interesting phenomenon about VAEs such as links between disentangled representations and robustness, and tradeoffs between disentanglement and reconstruction accuracy.
3. The method also provides some protection for downstream classification tasks.

Weaknesses:

1. There is a lack of baselines to compare against, and the paper has not really stress-tested the algorithm to ensure it is in fact robust. Most of the literature on adversarial examples does significantly more extensive testing to attempt to break the model.
2. I found the story and the experimental evaluation a bit incomplete. Though the experiments demonstrate that the method achieves success in practice, the paper does not seem to sufficiently explore "why" it seems to work. I think this is a bit important here considering the previous point regarding baselines and testing. At the moment, it seems like the paper combines these two ideas of learning disentangled representation and hierarchical VAEs in a somewhat ad-hoc manner which ends up providing some robustness, but it is a bit opaque why this is supposed to work. Some experiments or ablation studies which establish a more direct link between disentangled representations and robustness will be helpful.

Overall, I think this is a decent paper but I don't feel that I can advocate strongly for it since the scientific contribution seems a bit limited.

------Updates after author response------

I thank the authors for the response and the new experiments. In light of the clarifications and additional evaluations, I have increased my score to 7.

---

> ### Author Response · Authors · 2020-11-14
> **Response to Reviewer #3**
>
> Thank you for your thoughtful review.  We are glad you like the paper and thank you for highlighting the novelty of our work. We have uploaded a significant update to the paper to address the concerns raised, including a number of new experiments.
>
> ----------------------
>
> “1. There is a lack of baselines to compare against... Most of the literature on adversarial examples does significantly more extensive testing to attempt to break the model.“
>
>
> The lack of more baselines is a direct result of the novelty of this work. There is a dearth of established approaches for how to defend VAEs and we believe a real strength of the paper is in how we break new ground here, rather than just providing a “better version” of something that has already been done. We note that we do consider vanilla VAEs, hierarchical VAEs and β-TCVAEs as baselines, which we believe are representative of what most practitioners would consider using a present.
>
> As for the range of attacks considered, we agree that more attacks would increase the strength of the paper and have now added these (in Figure 5), but note that because the area is less developed than conventional adversarial settings, there are only a few known attacks in the literature. To this end, we have added two attacks to the paper. Firstly, we directly attack the output of the model, aiming to increase the ELBO for the target image under the attacked input - this attack is proposed in Kos et al. (2018). Secondly, we have added our own novel latent-space attack, where we aim to minimise the 2-Wasserstein distance between the target and attacked posterior distributions. In both these attacks Seatbelt-VAEs outperform β-TCVAEs and β-TCVAEs outperform vanilla VAEs. We also tried attacks proposed by Kos et al. that aim to reduce the L2 distance between samples in the latent space but found that these attacks were ineffective against all models considered, such that they provide no additional useful comparisons.
>
> ----------------------
>
> “2. I found the story and the experimental evaluation a bit incomplete...the paper does not seem to sufficiently explore "why" it seems to work... Some experiments or ablation studies which establish a more direct link between disentangled representations and robustness will be helpful."
>
>
> We think there may have been a bit of a confusion here: as explained in our general response to all reviewers, we are not trying to establish a direct link between disentangled representations and robustness, but instead that regularisers that were proposed for disentangling can be reliably used to confer robustness to VAEs.  As such, we did not include experiments trying to establish this direct link between disentanglement and robustness as we are not advocating that such a link exists.
>
> In fact, we have now added a new experiment in Appendix E.1 that suggests that there does not seem to be a clear connection between disentangling, as measured by MIG (Chen et al., 2018), and adversarial robustness.  We have also updated Section 3 to make the links between regularisation of the latent space and adversarial robustness clearer. Our methods are directly motivated by considerations of the amount of *overlap* in the aggregate posterior and how this can be best controlled while training the model. Controlling overlap is a necessary but not sufficient condition for achieving disentanglement (Mathieu et al., 2019), whereas it does seem sufficient for achieving robustness, hence the lack of a direct link between the two.
>
> More generally, we believe that the experimental evaluation in this paper is unusually thorough, particularly now we have added numerous additional experiments for this rebuttal process. We have trained 36 β-TCVAEs with varying latent spaces sizes and β values and 30 Seatbelt-VAEs of varying depths and β values for each dataset. Overall we performed ~15000 attacks per model per attack methodology for each dataset. We have more than a dozen different experimental evaluations, between the main paper and appendices, many of which are are ablation studies or experiments investigating why things work (e.g. Figure 2 [left] confirming overlap is indeed increased, ablations over different β, λ, and d_z throughout, ablations of different L in Appendix D, investigations of Appendix E, new considerations of different attacks).  Nonetheless, we are very open to suggestions if there are particular experiments you feel are missing.

---

### Author Response · Authors · 2020-11-14
**Overview Response to All Reviewers**

We thank reviewers for their thoughtful and insightful feedback. We were pleased to note that the novelty and relevance of the work were highlighted by all reviewers.  There is a real need for the development of adversarially robust VAEs and this work provides a first step in that direction.

We have run numerous new experiments: confirming quantitatively the more stark decrease in quality of reconstructions in β-VAEs than in β-TCVAEs; adding a new downstream task where convolutional neural networks are trained on image reconstructions; and also we have added to the paper two adversarial attacks, one of which is a novel attack method, to further stress-test the TC-penalised models.

Before describing these new experiments, we wished to clear up some confusion as to the connection between our work and that of disentangling. In this work, we are contending that objectives first proposed for encouraging disentanglement, not disentangled representations themselves, can create robust VAEs. We highlight that these have a regularising effect on a VAE’s encoding process, and we are exploiting this property, not the potentially disentangled representations, to induce robust VAEs. In short, we are suggesting inducing robustness by controlling the overlap of the representations, something that has shown to be a necessary but not sufficient condition for disentanglement (Mathieu et al 2019).  Any link to disentanglement itself is thus indirect.

In practice, these methods do not always directly induce disentangled representations. Indeed we have added a new experiment (in Appendix E.1) that shows that the MIG (a measure of disentanglement) is uncorrelated with our robustness measures in the case of β-TCVAEs. The reason for this is that while the regularisation was reliably able to induce robustness, achieving disentanglement is substantially more finicky and requires a combination of careful parameter tuning and luck: the setups we are using do not actually reliably produce disentangled representations, even though they do reliably produce robust ones.

To make this all clearer in the paper, we have also edited Section 3.1 to underline that we are not exploring the connection between disentangled representation and robustness, but rather are simply using TC-penalisation — because of its ability to regularise the VAE encoding process — to train robust VAEs.

We were also pleased that reviewers noted the breadth and quantity of our empirical evidence, which we have bolstered during this rebuttal period in three key ways (on top of the MIG comparisons above):

1)  We have added a subfigure in Figure 2 showing the value of E_{q(**z**|**x**)} [log p(**x**|**z**)] for β-VAEs and β-TCVAEs as a function of β. The value of this quantity decreases more quickly for β-VAEs than β-TCVAEs. This plot demonstrates quantitatively that VAE reconstructions are more robust to TC regularisation than to regularisation of KL(q(**z**|**x**)||p(**z**)).

2) We have added a new downstream task to Table 1, classification of image reconstructions using fully convolutional neural networks. Again β-TCVAEs are more robust to adversarial attack on this task than vanilla VAEs, and Seatbelt VAEs more robust still.

3) To further stress-test β-TCVAEs and Seatbelt-VAEs we have included results for two other forms of attack, one which uses the reconstruction of the VAE, aiming to increase the ELBO for the target image under the attacked input (as proposed by Kos et al. 2018) and a novel mode of attack where we use the 2-Wasserstein distance metric to attack the VAE latent space. We find that Seatbelt-VAEs also offer protection to these attacks, cementing our findings that these models are broadly adversarially robust. These results complement our original findings on the KL-based latent space attacks, which were found to be the most effective against VAEs by Gondim et al. 2018. All these results have been combined into Figure 5.

We think the paper has been significantly strengthened by these new experiments. Again we thank reviewers for their feedback, which stimulated these improvements.

---

### Decision · Program_Chairs · 2021-01-07
**Final Decision**

**Decision:**

Accept (Poster)

**Comment:**

This paper presents a hierarchical version of β-TCVAE that promotes disentanglement in the latent space and improves the robustness of VAEs over adversarial attacks, without (much) degeneration on the quality of reconstructions. The analysis on the relationship between disentanglement and adversarial robustness is valuable and the method is new. The results look promising. The comments were properly addressed.